# Robust enzyme discovery and engineering with deep learning using CataPro

Zechen Wang ®[1], Dongqi Xie[2], Dong Wu[2], Xiaozhou Luo ®[3,4,5], Sheng Wang[2], Yangyang Li ®[1], Yanmei Yang ®[6] ✉, Weifeng Li ®[1] ✉ & Liangzhen Zheng ®[2,7] ✉

Accurate prediction of enzyme kinetic parameters is crucial for enzyme exploration and modification. Existing models face the problem of either low accuracy or poor generalization ability due to overfitting. In this work, we first developed unbiased datasets to evaluate the actual performance of these methods and proposed a deep learning model, CataPro, based on pre-trained models and molecular fingerprints to predict turnover number ($k_{cat}$), Michaelis constant ($K_m$), and catalytic efficiency ($k_{cat}/K_m$). Compared with previous baseline models, CataPro demonstrates clearly enhanced accuracy and generalization ability on the unbiased datasets. In a representational enzyme mining project, by combining CataPro with traditional methods, we identified an enzyme (SsCSO) with 19.53 times increased activity compared to the initial enzyme (CSO2) and then successfully engineered it to improve its activity by 3.34 times. This reveals the high potential of CataPro as an effective tool for future enzyme discovery and modification.

Enzymes, as biocatalysts, play pivotal roles in industrial processes such as pharmaceutical and chemical manufacturing, food and beverage processing, and biofuel synthesis[1-8]. Natural crude enzymes from microorganisms, plants or animals have long been used in chemical synthesis in reactions such as hydration, dehydration and transglycosylation[9-11]. Whereas for non-natural reactions for pharmaceutical intermediates, enzymes can be screened or designed to exhibit broad substrate specificity and high enantioselectivity in biochemical experiments[12]. For example, transaminases are commonly used in the biosynthesis of chiral amines with quite high enantioselectivity[13,14]. With the assistance of computational methods, enzymes can be designed to catalyze non-native reactions, such as the Diels-Alder reaction, exhibiting high stereoselectivity for carbon-carbon bond formation[15], and the efficiency of the de novo-designed enzymes may be comparable to that of natural enzymes if careful computational scoring filters are applied[16-18]. Alternatively, a more efficient strategy for enzyme discovery is based on the deep learning (DL)-based protein structural prediction, structural similarity calculation and clustering of previously uncharacterized proteins from vast protein sequence databases[19-21]. In a study by Huang et al., the authors described an AlphaFold2-based structure prediction[22] and clustering protocol for deaminase discovery and further designed a smallest single-strand-specific cytidine deaminase[19]. Similarly, three polyethylene terephthalate degradation enzymes were identified and analyzed using a structure-based clustering strategy from ocean marine genomics data[20]. Although these structure-based DL methods have proven effective for enzyme discovery, they may exhibit some randomness and chance when applied to certain specific substrates, particularly non-natural substrates[23,24]. Therefore, we hypothesize that combining substrate-specific enzyme activity prediction models with protein structure clustering-based enzyme discovery strategies could enhance both efficiency and specificity.

[1]School of Physics, Shandong University, Jinan 250100 Shandong, China. [2]Shanghai Zelixir Biotech Co. Ltd, Shanghai 201210 Shanghai, China. [3]Shenzhen Key Laboratory for the Intelligent Microbial Manufacturing of Medicines, Shenzhen Institute of Advanced Technology, Chinese Academy of Sciences, Shenzhen 518055 Guangdong, China. [4]Key Laboratory of Quantitative Synthetic Biology, Shenzhen Institute of Advanced Technology, Chinese Academy of Sciences, Shenzhen 518055 Guangdong, China. [5]Center for Synthetic Biochemistry, Shenzhen Institute of Synthetic Biology, Shenzhen Institute of Advanced Technology, Chinese Academy of Sciences, Shenzhen 518055 Guangdong, China. [6]College of Chemistry, Chemical Engineering and Materials Science, Key Laboratory of Molecular and Nano Probes, Ministry of Education, Shandong Normal University, Jinan 250014 Shandong, China. [7]Shenzhen Zelixir Biotech Co. Ltd, Shenzhen 518107 Guangdong, China. ✉e-mail: yym@sdnu.edu.cn; lwf@sdu.edu.cn; zhenglz@zelixir.com

Meanwhile, wild-type enzymes often do not meet the rigorous demands of industrial production, necessitating enhancements through the introduction of single or multiple mutations[25]. To this end, enzyme evolution relies on experimental methodologies that facilitate the engineering of enzymes with enhanced properties. Directed evolution is a laboratory-based method that simulates natural evolution to enhance enzyme characteristics[25,26]. This iterative process involves generating genetic diversity through random mutagenesis or DNA shuffling, followed by screening or selection for variants exhibiting desired traits[27]. Although many enzymes with good properties have been developed via directed evolution, it is a time-consuming and cost-ineffective process.

Computational rational enzyme design, by contrast, offers an alternative option for finding favorable mutations within limited experimental trials[15,28,29]. Several commonly employed computational strategies are pivotal for enhancing enzyme functionality. Researchers rely on co-evolution or conservation-based analysis by multiple sequence alignments to locate highly variable residues, and identify key residues for substrate specificity using molecular docking, locate the flexible loops through structure prediction and molecular simulations, and predict the residue mutation-induced protein thermostability changes[30,31]. These computational methods offer valuable insights for determining the functional changes in enzymes induced by mutations, underscoring the importance and promising application potential of developing accurate and reliable tools to advance enzyme discovery and engineering.

Currently, numerous entries describing in vitro enzyme catalytic efficiencies (kinetic parameters) have been recorded in various open-source databases, such as BRENDA[32] and SABIO-RK[33], establishing a foundation for developing enzyme kinetic parameter prediction algorithms. The main kinetic parameters include the maximum turnover number ($k_{cat}$) and Michaelis constant ($K_m$), which define the maximum chemical conversion rate of a reaction and the substrate concentration when the enzyme reaches half of its maximal conversion rate, respectively, while $k_{cat}/K_m$ represents the catalytic efficiency of the enzyme[34]. These entries provide valuable data for designing algorithms that predict enzyme kinetic parameters. Computational techniques in bioinformatics have undergone a revolution with the development of machine learning (ML) and DL algorithms[35–37]. Several methods for predicting enzyme kinetic parameters have been reported. Heckmann et al. successfully used ML with integrated features to predict $k_{cat}$ values for enzymes in *Escherichia coli*, both in vivo and in vitro[38]. Kroll et al. developed an organism-independent DL model that predicts $K_m$ values for natural enzyme-substrate combinations[39]. Later, Kroll et al. improved predictions of kinase inhibition and enzyme-substrate relationships by using a multimodal Transformer network[40]. A DL-based predictor, DLKcat, has been developed for high-throughput $k_{cat}$ prediction for enzymes from any organism, using substrate structures and enzyme sequences as inputs[41]. However, the relatively simple encoding of protein sequence by DLKcat may not be suitable or effective when there is limited data. Subsequently, as protein language models have advanced, utilizing embeddings from these models to represent protein information has become a prevalent approach, demonstrating superior performance in various downstream tasks. For instance, TurNup utilized the fine-tuned esm-1b vectors and differential reaction fingerprints as representations of enzyme-catalyzed reactions to predict the $k_{cat}$ value of natural reactions of wild-type enzymes[42]. UniKP[43] and MPEK[44] employed Prot_t5_xl_uniref50 to generate enzyme features and attempted to incorporate environmental factors as inputs for predicting enzyme kinetic parameters. However, they do not systematically examine the effects of diverse enzyme and substrate representations on enzyme kinetic parameter prediction. Furthermore, a small amount of environmental information may not allow the model to learn the impact of environmental factors.

Although these models demonstrate promising performance in certain test scenarios, they still have some limitations. The aforementioned studies generally combine data from various reactions involving natural enzymes as well as mutations, and then randomly split them into training and testing sets[41,43]. However, this approach may lead to data leakage. In many bioinformatics tasks, such as protein function prediction[45–48] and protein-ligand binding affinity prediction[49–52], the high sequence similarity between proteins in the training and test sets can lead to overly optimistic evaluations and systemic bias in the models. The currently available enzyme kinetic parameter data represents only the tip of the iceberg of all chemical reactions occurring in nature. Therefore, developing models based on existing data that are applicable across a wide range of enzymatic reaction systems is one of the greatest challenges in current enzyme kinetic parameter prediction.

It is generally hypothesized that enzyme catalytic abilities are highly evolutionary-driven, especially with their naive substrates. This hypothesis makes protein language models naturally suitable for enzyme catalytic predictions. Protein language models provide an embedding with limited dimensions that contain denser information than residue one-hot encoding. Furthermore, the embedding generally is generally highly correlated to residue probabilities and co-evolutions, and is commonly used for predicting protein-protein interaction and protein annotation prediction[53–57]. In addition, different pre-trained protein language models have different properties. Consequently, it is interesting to know which one is most suitable for robust enzyme catalysis prediction.

To address the aforementioned issues, we established a benchmark to evaluate models for predicting enzyme kinetic parameters ($k_{cat}$, $K_m$, and $k_{cat}/K_m$). We collected the latest enzyme-substrate entries containing either $k_{cat}$ or $K_m$ values from the BRENDA[32] and SABIO-RK[33] databases to construct the initial $k_{cat}$, $K_m$, and $k_{cat}/K_m$ datasets. For each dataset, entries were clustered based on a protein sequence similarity threshold of 0.4. These clusters were divided into ten partitions, thereby creating an unbiased dataset for ten-fold cross-validation. We then constructed an enzyme kinetic parameter prediction framework called CataPro, based on the embeddings of pretrained language models and molecular fingerprints. All training and testing were conducted on unbiased datasets to ensure fair comparison. The results indicate that CataPro has superior accuracy and generalization ability compared to other baseline models in predicting $k_{cat}$, $K_m$, or $k_{cat}/K_m$. Additionally, we also evaluated the ability of CataPro and other baseline kinetic parameter models to identify enzyme mutation effects using the ten-fold unbiased cross-validation datasets. To further assess the ability of CataPro to rank mutants, we also assessed it using deep mutational scanning (DMS) data[58] and several small experimental datasets from the literature, which demonstrated that CataPro has considerable potential for enzyme engineering. Finally, CataPro was used to assist in exploring enzymes catalyzing the conversion of 4-vinylguaiacol (4-VG) to vanillin, resulting in the discovery of a highly active alternative enzyme *Sphingobium sp*. CSO (SsCSO) with an activity 19.53 times that of the initial enzyme. After further sequence optimization with CataPro, the activity of a high-activity mutant was increased 3.34-fold compared to the original SsCSO. These experiments demonstrate the practical application value of CataPro in enzyme discovery and directed evolution.

## Results

### Overview of CataPro

During the dataset preparation phase, we first created the initial $k_{cat}$ and $K_m$ datasets by collecting all $k_{cat}$ and $K_m$ entries, respectively, from the BRENDA[32] and SABIO-RK[33] databases. Details of the $k_{cat}$ and $K_m$ data collection and processing can be found in Supplementary Figs. 1–2, and the analysis of processed $k_{cat}$ and $K_m$ data is shown in Supplementary Fig. 3. Samples containing both experimental $k_{cat}$ and $K_m$

values were extracted to construct the $k_{cat}/K_m$ dataset. To establish a fair benchmark for evaluating the accuracy and generalization capability of enzyme kinetic parameter prediction models, we applied CD-HIT[59] based sequence clustering (sequence similarity cutoff is 0.4) within each dataset to form ten enzyme groups. Enzyme sequences and canonical SMILES of substrates were collected from UniProt[60] and PubChem[61], respectively. Thus, the unbiased ten-fold cross-validation datasets for $k_{cat}$, $K_m$, and $k_{cat}/K_m$ were created (Fig. 1a).

The CataPro model is a neural network-based framework for predicting enzyme kinetic parameters, using amino acid sequences and SMILES as the original representations for enzymes and substrates, respectively. Enzyme information is encoded into a vector having a length of 1024 using the ProtT5-XL-UniRef50 model (referred to as ProtT5)[57], similar to UniKP. For substrates, we employed MolT5 embeddings[62] and MACCS keys fingerprints jointly as feature representations, having dimensions of 768 and 167, respectively. The enzyme and substrate representations are concatenated into a 1959-dimensional vector, serving as the overall information for the enzyme-catalyzed reaction. This concatenated vector is used as the input to the neural network for predicting the $k_{cat}$ or $K_m$ values.

For predicting $k_{cat}/K_m$, some researcher employed the same architecture used for predicting $k_{cat}$ and $K_m$[43]. However, this approach may not achieve optimal performance, as it makes $k_{cat}/K_m$ entirely independent of $k_{cat}$ and $K_m$. To mitigate this concern, we proposed a strategy that incorporates a neural network-based correction term to enhance the accuracy of $k_{cat}/K_m$ predictions. Firstly, the $k_{cat}$ and $K_m$ models were pre-trained using the unbiased $k_{cat}/K_m$ dataset with ten-fold cross-validation. Then, the initial $k_{cat}/K_m$ predictions were derived by dividing the predictions from the pre-trained $k_{cat}$ models by those from the pre-trained $K_m$ models. Due to the inherent errors in the $k_{cat}$ and $K_m$ models, the initial $k_{cat}/K_m$ predictions may exhibit substantial inaccuracies. Therefore, a correction model was trained to mitigate this error. The input to the correction term prediction network

remains a 1959-dimensional vector representing the enzyme-substrate feature, which is transformed into a 256-dimensional vector after passing through a dense layer. Then, a feature-wise attention layer before the output layer extracts crucial information. The predicted correction, combined with the ratio of $k_{cat}$ to $K_m$, yields the final predicted $k_{cat}/K_m$ value. During the training phase of the correction factor model, the parameters of the pre-trained $k_{cat}$ and $K_m$ models are held fixed.

## Performance of CataPro in $k_{cat}$ and $K_m$ predictions

In this work, unless otherwise stated, all training and testing were conducted on our unbiased ten-fold cross-validation datasets. Evaluation metrics include Pearson's correlation coefficient (PCC), Root-Mean-Square-Error (RMSE), and Spearman's correlation coefficient (SCC). The inclusion of SCC is particularly important as ranking power is crucial in enzyme mining and enzyme engineering. We retrained DLKcat and UniKP on our datasets under the same testing conditions to ensure fair comparison. Other enzyme and substrate representations were also evaluated, including embeddings from the protein language models like ESM2 (esm2_t33_650M_UR50D)[63] and SaProt (SaProt_650M_AF2)[55], embeddings from chemical molecule language model such as Mole-BERT[64], and traditional rule-based representations such as Morgan fingerprints and RDKit fingerprints.

With the unbiased dataset of $k_{cat}$, CataPro achieved PCC, SCC, and RMSE values of 0.497, 0.495, and 1.329, respectively, which are significantly better than those of DLKcat and UniKP (Fig. 2a–c and Supplementary Fig. 4a). For the $K_m$ prediction, the performance of most models was comparable (Fig. 2d–f and Supplementary Fig. 4b). The PCC, SCC, and RMSE achieved by CataPro are 0.633, 0.629, and 0.998, respectively, slightly better than UniKP. These results indicate that predicting $k_{cat}$ is more challenging than predicting $K_m$. This is because $k_{cat}$ represents the maximum number of catalytic reactions an enzyme can catalyze per unit time, encompassing the entire catalytic process,

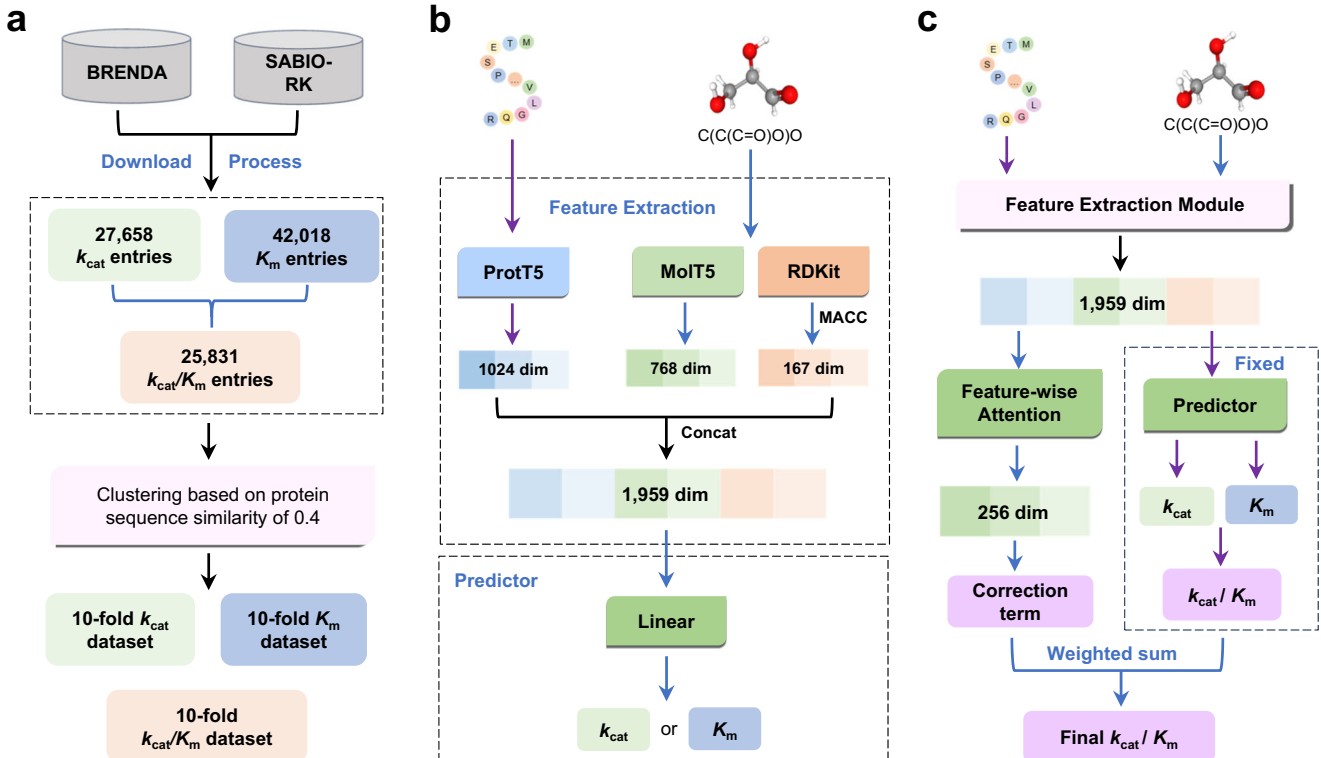

**Fig. 1 | Overview of CataPro. a** Establishment of ten-fold unbiased dataset for $k_{cat}$, $K_m$, and $k_{cat}/K_m$. The enzyme sequence similarity between each fold data is lower than 0.4. **b** Kinetic parameter ($k_{cat}$ or $K_m$) prediction framework based on language model embeddings and molecular fingerprints. **c** $k_{cat}/K_m$ prediction framework based on $k_{cat}$ and $K_m$ pre-trained models.

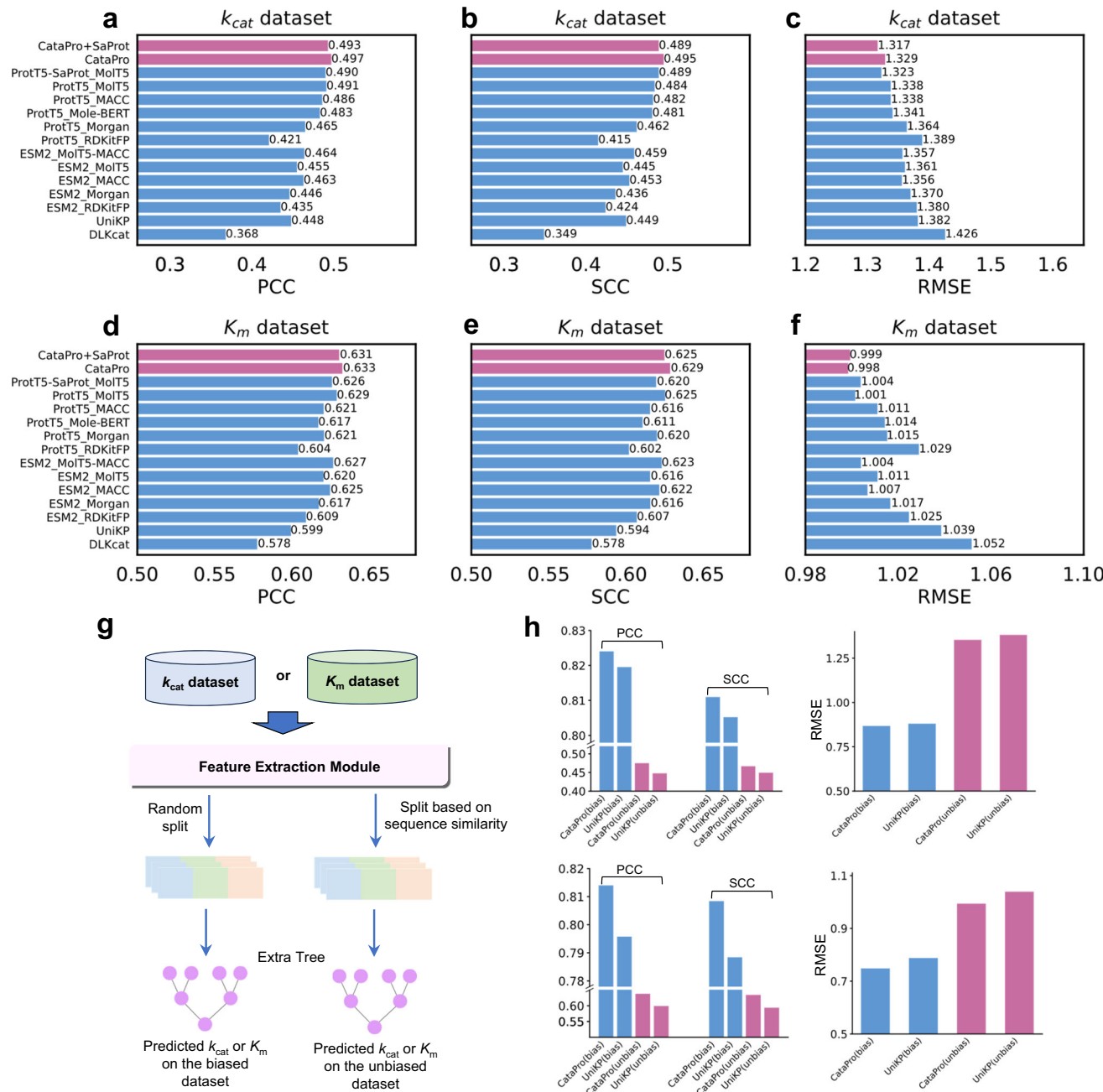

**Fig. 2 | Performance comparison of various $k_{cat}$ and $K_m$ models on 10-fold unbiased datasets. a, b**, and **c** show the PCC, SCC, and RMSE achieved by the $k_{cat}$ prediction models on the $k_{cat}$ dataset. **d, e**, and **f** show the PCC, SCC, and RMSE achieved by the $K_m$ prediction models on the $K_m$ dataset. It is worth noting that all models, including DLKcat and UniKP, were trained (or re-trained) on the 10-fold unbiased datasets for $k_{cat}$ and $K_m$. In panels **a-f**, CataPro and CataPro+SaProt are highlighted in red, while other models are represented in blue. **g** Workflow of Extra Tree-based CataPro. **h** shows the performance of CataPro (Extra Tree) and UniKP on unbiased and biased datasets. The top two subplots in panel **h** display the results for the $k_{cat}$ dataset, while the bottom two subplots show the results for the $K_m$ dataset. In the four subplots of panel **h**, the performance of models on the unbiased datasets is highlighted in red, while the performance on the biased datasets is represented in blue. Source data are provided as a Source Data file.

including transition state formation, overcoming energy barriers, and product release, while $K_m$ is an inherent property of the enzyme towards the substrate, related to the strength of the enzyme-substrate binding. This further demonstrates that processes involving complex reaction mechanisms are indeed challenging to model directly using only enzyme sequences and substrate structures.

In addition, an interesting phenomenon is that embeddings from ProtT5[57] are more effective than those from ESM2[63] in $k_{cat}$ prediction tasks. In terms of substrate representation, MolT5 embeddings, Mole-BERT embeddings, and MACCS keys fingerprints exhibit similar

effects. Furthermore, we explored the effect of protein structural features on enzyme kinetic parameter prediction. SaProt[55], trained on protein 3D structure, was employed to derive structural characteristics from enzyme structure predicted by AlphaFold2. After incorporating structural features, the model in the $k_{cat}$ prediction task was slightly worse, while RMSE decreased from 1.329 to 1.317. However, in the $K_m$ prediction task, structural features did not have a significant impact. Another study has also reported the ineffectiveness of structural features in predicting enzyme kinetic parameters[65]. This implies there is still considerable room for improvement in structure-based protein

pre-trained models. Finally, we separately present the results achieved by models for $k_{cat}$ and $K_m$ prediction on wild-type and mutant enzymes, demonstrating that CataPro still maintains substantial advantages over DLKcat and UniKP (Supplementary Figs. 5–6). A recent study reported a model called ProSmith for $K_m$ prediction, which achieved a determination coefficient ($R^2$) of 0.41 on a subset where the maximum enzyme sequence identity with the training set was below 30%, demonstrating a strong generalization ability[40]. The similar performance of ProSmith and CataPro in $K_m$ prediction may be attributed to both models utilizing transformer-based pretrained models to capture enzyme-substrate interactions, as well as the same data sources (BRENDA and SABIO-RK) for the datasets.

Han et al.[43] states that, extra trees are more effective than neural networks in predicting enzyme kinetic parameters. We tested CataPro on the dataset collected by Li et al.[41] and found that the extra tree method indeed performed better, especially for the UniKP model (Supplementary Fig. 7). We also conducted additional testing of CataPro and other models on randomly-split ten-fold cross-validation $k_{cat}$ and $K_m$ datasets we developed. For this assessment, ten subsets of each dataset contained homologous enzyme data, a scenario similar to most testing setups reported in literature[41,43,44]. The results demonstrate that UniKP outperformed most models in this testing scenario, especially in the $k_{cat}$ prediction task (Supplementary Fig. 8). Consequently, we employed extra trees algorithm from scikit-learn package to learn the relationship between the features of CataPro and experimental $k_{cat}$ and $K_m$ values (Fig. 2g). As anticipated, CataPro trained with extra trees achieved optimal results (Supplementary Fig. 8). We further compared the performance of CataPro (extra tree) and UniKP on unbiased and biased datasets, and found that these two data partitioning methods result in noteworthy differences in model performance, while the two models perform similarly when evaluated on the same dataset (Fig. 2h). This further suggests that random dataset splitting can lead to overly optimistic performance for the model, as enzymes in real-world applications are often likely to be unfamiliar to it. Therefore, the primary advantage of our proposed CataPro lies in utilizing unbiased datasets for training and evaluation, rather than relying on a more sophisticated model architecture.

Given that many enzymatic reactions may involve multiple substrates, we examined the applicability of CataPro to multi-substrate reactions. Recently, Kroll et al. curated a high-quality $k_{cat}$ dataset for wild-type enzyme, which includes enzyme sequences, substrate IDs, reaction equations, and kcat measurements. For each reaction in the test set, we used CataPro to predict the $k_{cat}$ values for the enzyme in combination with each substrate, and the average of these $k_{cat}$ values was taken as the final predicted $k_{cat}$ value for the reaction. CataPro achieved a PCC of 0.661 and an $R^2$ of 0.415 on this test set, which is very close to TurNuP's performance with a PCC of 0.67 and an $R^2$ of 0.44. This suggests that CataPro also has the potential to handle multi-substrate reactions. Following the method of Kroll et al., we partitioned the test set into four subsets based on the maximal sequence identity to enzymes in the CataPro $k_{cat}$ dataset. The maximum sequence identity intervals for these subsets were 0–40%, 40–80%, 80–99%, and 99–100%. The performance of CataPro in each subset is presented in detail in the Supplementary Fig. 9a. Notably, in the 0–40% identity subset, CataPro achieved a PCC of 0.410 and an $R^2$ of 0.133, showing a considerable gap compared to TurNuP's $R^2$ of 0.33 in this subset. This suggests that CataPro may still face challenges when dealing with unfamiliar reactions involving multiple substrates, underscoring the importance of evaluating the model in a broader range of application scenarios to avoid overly optimistic results that may arise from relying solely on the 10-fold cross-validation dataset. Additionally, we retrained CataPro on the training set curated by Kroll et al., which contains 3,421 kcat data points, to further explore the potential of the CataPro architecture for tasks involving multi-substrate reactions. For reactions with multiple substrates, the final

substrate features were represented by the average of the features of all substrates. We trained CataPro using 5-fold cross-validation, consistent with the folds used by Kroll et al. During inference, the final prediction was obtained by averaging the predictions from the five models. On the TurNuP test set, the retrained CataPro achieved a PCC of 0.672, an $R^2$ of 0.451, and a MSE of 0.783, which are highly consistent with TurNuP's PCC of 0.67, $R^2$ of 0.44, and MSE of 0.81. In the four subsets divided based on the maximum sequence identity to enzyme sequences in the training set, the retrained CataPro also achieved performance similar to TurNup, with an $R^2$ of 0.333 on the 0–40% subset (Supplementary Fig. 9b). This suggests that when the CataPro architecture is specifically trained on datasets involving multi-substrate reactions, it can achieve performance comparable to the current state-of-the-art (SOTA) models. This provides insights for the future development of enzyme kinetic parameter prediction models utilizing multi-substrate inputs.

## Transfer learning improves $k_{cat}/K_m$ prediction

In this study, the $k_{cat}/K_m$ dataset consists of enzyme-substrate pairs containing $k_{cat}$ and $K_m$ values, and the distribution of samples across various $k_{cat}/K_m$ ranges is shown in Fig. 3a. These samples are stratified into ten folds based on a sequence similarity threshold of 0.4, with label distributions illustrated in Fig. 3b.

Although $k_{cat}/K_m$ can be directly calculated from the predicted $k_{cat}$ and $K_m$, this may not be reasonable because errors in the $k_{cat}$ and $K_m$ values can potentially amplify errors in the $k_{cat}/K_m$ values. A better approach is to directly use the model to learn $k_{cat}/K_m$ values, as exemplified by UniKP. However, this neglects the relationship between the $k_{cat}/K_m$ value and the individual $k_{cat}$ and $K_m$ values. To overcome these challenges, CataPro adopts an architecture that integrates pre-trained models for $k_{cat}$ and $K_m$ prediction with a neural network-based correction term to predict $k_{cat}/K_m$. On the unbiased datasets of $k_{cat}/K_m$, CataPro achieved a PCC of 0.413, SCC of 0.416, and RMSE of 1.619, significantly outperforming DLKcat and UniKP (Fig. 3c–f). The ProT5_molT5-MACC shown in Fig. 3d–f demonstrates the performance when directly using the architecture shown in Fig. 1b to train $k_{cat}/K_m$. Comparison between CataPro and ProT5_molT5-MACC demonstrates that the architecture implemented in CataPro results in an increase in PCC from 0.392 to 0.413 and a decrease in RMSE from 1.64 to 1.619. This showcases the effectiveness of the feature combination and architecture adopted by CataPro in predicting $k_{cat}/K_m$. The models was also tested on the randomly partitioned ten-fold cross-validation datasets of $k_{cat}/K_m$, confirming the same conclusions as observed in the previous $k_{cat}$ and $K_m$ tests (Supplementary Fig. 10).

## CataPro enhances the ranking ability for mutations

In industrial production, optimizing enzymatic catalytic efficiency is crucial for reducing production costs. However, the inherent activity of natural enzymes often falls short of the requirements for high catalytic performance. Consequently, enhancing the activity of natural enzymes through modification has consistently been a primary objective within the enzyme industry. Despite the widespread acknowledgment of this challenge, recent research on predicting kinetic parameters has rarely provided a systematic examination of the models' capacity to differentiate between mutants in specific catalytic reactions. This study introduces an evaluation metric aimed at assessing the efficacy of enzyme kinetic parameter prediction models in the context of enzyme engineering, underscoring the importance of accurately predicting mutation effects for successful enzyme optimization.

We first selected reactions from each dataset ($k_{cat}$, $K_m$, and $k_{cat}/K_m$ datasets) where the number of mutants, including the wild-type exceeded a certain minimum threshold $N$. The criterion for defining the same reaction was based on identical UniProtID-SMILES pairs for enzyme and substrate. During training on the ten-fold cross-validation

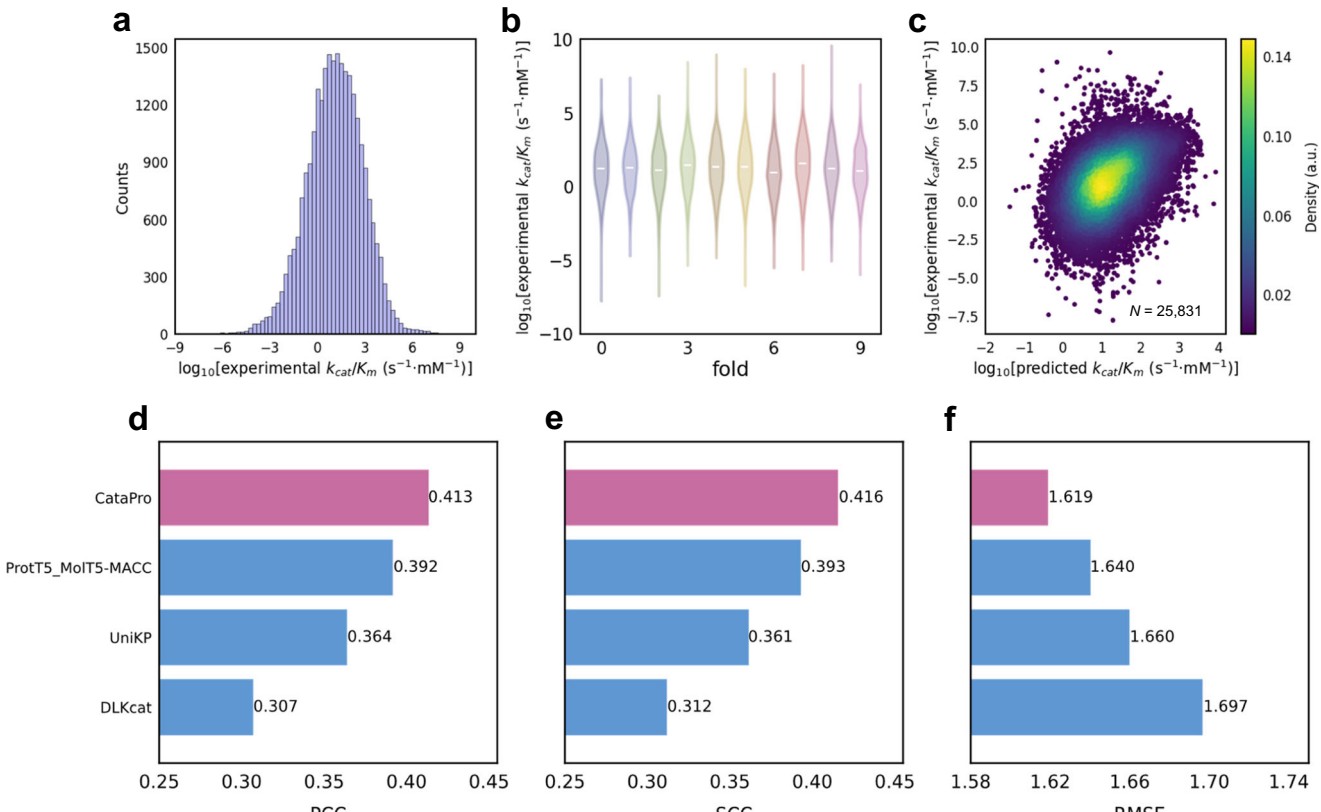

**Fig. 3 | Performance comparison of $k_{cat}/K_m$ models. a** Distribution of $k_{cat}/K_m$ values in the $k_{cat}/K_m$ dataset. The $k_{cat}/K_m$ dataset consists of samples with concurrent $k_{cat}$ and $K_m$ entries, which are divided into ten groups based on enzyme sequence similarity of 0.4. **b** Distribution of experimental $k_{cat}/K_m$ values in each fold of the ten-fold unbiased dataset is shown in different colors, with the white line in the body of the violin plot representing the median. Fold 0 contains 2,584 data points, while each of other nine folds contains 2,583 data points. **c** Performance of CataPro on the ten-fold unbiased dataset of $k_{cat}/K_m$, with the colorbar representing data density. **d, e**, and **f** respectively show the PCC, SCC, and RMSE achieved by the $k_{cat}/K_m$ prediction models. CataPro is highlighted in red in panels **d-f**. Source data are provided as a Source Data file.

datasets, predictions for the wild-type and mutants of selected reactions in the validation set were used for evaluation. Based on the distribution of the number of mutants within a single reaction in the current datasets, we evaluated the cases of $N$ being 20 and 30, respectively, with the corresponding number of reactions shown in Supplementary Figs. 11a-c. Most mutants involved in reactions with $N \geq 20$ and $N \geq 30$ across the three datasets are disadvantageous, showing worse effects than the wild-type (Supplementary Figs. 11d–f). For each reaction, the SCC between predicted and experimental values for all mutants was calculated as a measure of the model's ability to rank mutants within that reaction. The average SCC value achieved by the model across all reactions reflects its overall performance in ranking mutants. Figure 4a–c demonstrate that CataPro exhibits superior ranking ability compared to UniKP, particularly in the $k_{cat}$ and $k_{cat}/K_m$ datasets (Supplementary Table 1). We also tested the accuracy of the models in selecting the more favorable mutant between any two mutants (including the wild-type) within the same reaction. If a reaction contains $N$ mutants (including the wild-type), there are a total of $N(N\text{-}1)/2$ comparisons. According to Fig. 4d–f and Supplementary Table 2, CataPro still shows a clear overall advantage over UniKP and DLKcat.

Additionally, we adopted the method proposed by Kroll et al. to evaluate the models' ability to predict mutation effects[66]. Specifically, for each enzyme-substrate pair with $N \geq 20$ and $N \geq 30$, the mean of the measured values ($k_{cat}$, $K_m$, or $k_{cat}/K_m$) for the wild-type and all mutants was first calculated. The experimental and predicted values of each mutant and the wild-type in this reaction were then subtracted by the mean to obtain the experimental mutation effect and

the predicted mutation effect, respectively. Supplementary Figs. 12a–c illustrates that CataPro exhibits excellent correlation between predicted and experimental mutation effects for certain reactions, despite the training set containing no enzymes with more than 40% similarity to the enzyme. Conversely, for some reactions, CataPro demonstrates weaker correlations (Supplementary Figs. 12d–f). Notably, even for reactions with high correlation, CataPro demonstrates limitations in accurately predicting the absolute values of mutation effects. To perform a comprehensive evaluation of the model's ability to predict mutation effect, we concatenated the predicted and experimental mutation effect across all reactions. Unfortunately, in this test, none of CataPro, UniKP, or DLKcat successfully predicted the mutation effects (Supplementary Table 3). This may indicate that precisely capturing the effects of mutations remains a considerable challenge.

## Performance of CataPro on external test datasets from diverse sources

To further evaluate the potential of our model in enzyme mining and engineering, we collected four experimentally measured datasets from previously reported studies as additional external test sets. The first two test sets are the Tyrosine ammonia-lyase (TAL) homologue dataset and the TAL engineering dataset, collected from UniKP[43], both containing experimentally measured $k_{cat}/K_m$ values. TAL enzymes are utilized in the production of aromatic compounds, such as flavonoids, cinnamoyl anthranilates, or plastic precursors[67], and identifying their high-activity alternative enzymes and mutants has garnered considerable interest from the community. In a recent study, MPEK also

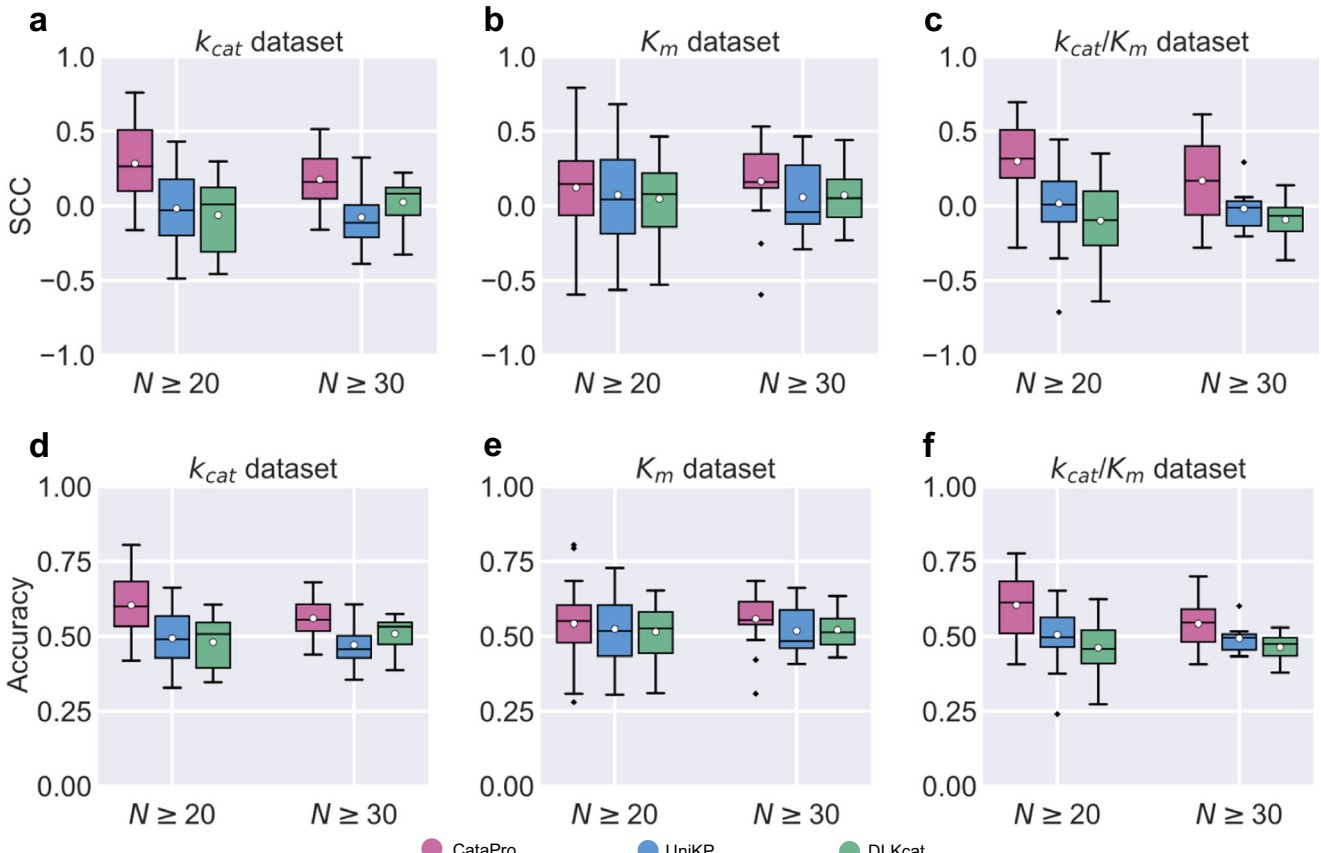

**Fig. 4 | Performance comparison of CataPro and UniKP in enzyme engineering scenarios. a-c** illustrate the models' ability to rank mutants within a specific reaction on the $k_{cat}$, $K_m$, and $k_{cat}/K_m$ datasets. To ensure statistical significance of the results, only enzyme-substrate reactions with a number of mutants (including the wild-type) exceeding $N$ (where $N$ is 20 or 30) in each dataset were used for this test. **d-f** show the accuracy of the models in identifying the better-performing mutants from any two mutants across these three datasets. The box plots depict the distribution of metrics achieved by the model across all reactions where the number of mutants is greater than or equal to $N$. In each box plot, the lower and upper boundaries of the box represent the first quartile (Q1) and the third quartile (Q3), respectively. The whiskers extend from the quartiles to the minimum and maximum values within 1.5 times the interquartile range. The white circle represents the mean value of each statistic and the black line inside the box represents the median. In the $k_{cat}$, $K_m$, and $k_{cat}/K_m$ datasets, the number of reactions with mutants (including the wild-type) where $N \geq 20$ are 39, 57, and 30, respectively, while the number of reactions with $N \geq 30$ are 10, 16, and 7, respectively. Source data are provided as a Source Data file.

used these two datasets for testing[44]. In the TAL homologue dataset, IsTAL has the lowest experimental $k_{cat}/K_m$ value. Taking IsTAL as a reference, if the model predicts higher $k_{cat}/K_m$ values for other enzymes than that of IsTAL, the prediction is considered correct (marked in green in Fig. 5a); otherwise, it is considered incorrect (marked in orange in Fig. 5a). It should be noted that in these two test datasets, except for a five-site mutant of RgTAL present in our $k_{cat}/K_m$ dataset, the maximum sequence similarities of all other test sequences to those in the $k_{cat}/K_m$ dataset were less than 0.67. The results in Fig. 5a indicate that CataPro has a 50% success rate in identifying TAL enzymes with higher activity than IsTAL. The TAL engineering dataset includes the wild-type RgTAL and its nine mutants. Among them, MT-587V, MT-10Y, and MT-489T exhibit enhanced catalytic efficiency compared to the wild-type, whereas the remaining five mutants demonstrate diminished catalytic efficiency. The results depicted in Fig. 5b illustrate that successful predictions were made for eight out of nine mutants, using the wild-type as a reference.

The other two small datasets involve the activity of D-2-deoxyribose-5-phos-phate aldolase (DERA) from *Escherichia coli* and BH1352 from *Bacillus halodurans* in catalyzing the D-2-deoxyribose-5-phosphate (DRP) reaction[68,69]. Catalyzing the DRP reaction is a crucial biocatalytic step in the conversion of renewable raw materials into valuable chemicals such as non-natural diol-1,3-BDO, which is used in the synthesis of polymers, pheromones, fragrances, insecticides, and

antibiotics. In the DERA dataset, all mutants exhibited lower activity than the wild-type. Interestingly, CataPro was able to identify the decrease in enzyme activity resulting from these mutations (Fig. 5c), despite the presence of five reaction data points for DERA catalyzing other substrates in the $k_{cat}/K_m$ dataset. The BH1352 dataset consists of nine single-point mutants and one double-point mutant, with the BH1352 enzyme exhibits a maximum sequence identity of only 0.54 to the enzymes in our $k_{cat}/K_m$ dataset. With the BH1352 dataset, CataPro achieved an 80% prediction success rate (Fig. 5d). These results indicate that CataPro exhibits robust performance in datasets from diverse sources, once again confirming its strong potential for practical applications.

**Performance of CataPro on deep mutation scanning datasets**

Traditional experimental methods usually assess only dozens of mutations at a time, which is trivial compared to the entire mutational landscape of the protein. Deep mutational scanning (DMS), an approach that integrates genotype and phenotype, enables the evaluation of fitness in up to one million protein variants in a single experiment[58]. While enzyme kinetic parameters are not fully correlated with fitness, which is influenced by diverse factors such as stability, folding efficiency, catalytic activity, binding specificity, and affinity[70], evaluating enzyme kinetic parameter models on the genotype-phenotype datasets remains meaningful.

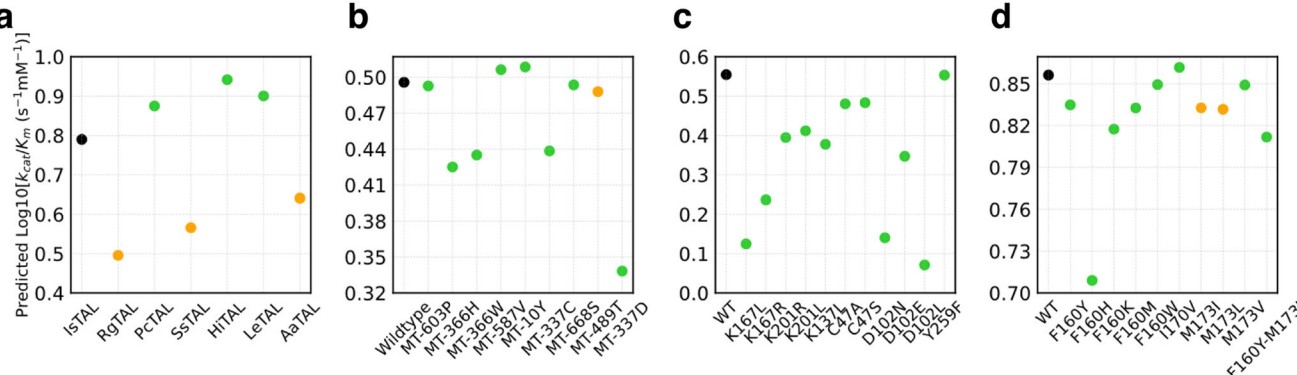

**Fig. 5 | Performance of CataPro on the small datasets measured in previously reported experiments. a-d** Predicted $k_{cat}/K_m$ values on the TAL homologue dataset, the TAL engineering dataset, the DERA dataset, and the BH1352 dataset, respectively. The black dots represent reference, while the green and orange dots denote correctly and incorrectly predicted samples, respectively. In other words, if CataPro successfully predicts that the activity of a specific sequence or mutant is higher or lower than that of the reference, it is marked in green; otherwise, it is marked in orange. Source data are provided as a Source Data file.

To evaluate the performance of the predictors on mutants across a broader range of sites, we collected DMS data for four enzymes (with their corresponding substrates) for testing. The first dataset is the EcTL dataset, which includes the $TEM-1\beta$-lactamase expressed in *Escherichia coli* (Fig. 6a). This enzyme hydrolyzes $\beta$-lactam antibiotics, such as ampicillin, thus conferring antibiotic resistance to bacteria[71]. This dataset contains fitness data for 5,468 mutants across 286 sites. The other three datasets involve SsIGPS from *Sulfolobus solfataricus*, TmIGPS from *Thermotoga maritima*, and TtIGPS from *Thermus thermophilus*, which are indole-3-glycerol phosphate synthases (IGPS) from diverse organisms[72] (Fig. 6b–d). Despite their similar functions, these IGPS enzymes exhibit low sequence similarity. Each of these IGPS enzymes has fitness data for approximately 1,513 mutants involving 80 sites.

We evaluated the performance of CataPro, UniKP, and DLKcat on these four DMS datasets, as these datasets encompass mutations at various sites either proximal or distal to the catalytic sites. This extensive mutation space better reveals the real performance of the models. In addition, the Position-Specific Scoring Matrix (PSSM) was used as an additional mutation-function prediction baseline. PSSM incorporates protein sequence co-evolution information, with each element representing the frequency of a specific amino acid occurring at a particular position across different homologous sequences. The conservation and mutation information encapsulated in the PSSM are frequently utilized as descriptors for protein engineering[73]. Interestingly, the SCCs achieved by the $k_{cat}$ and $k_{cat}/K_m$ models of CataPro either approach or surpass those obtained using PSSM scores (Fig. 6e–h). For UniKP and DLKcat, the performance of the $k_{cat}$ model of UniKP is relatively better, particularly on the EcTL dataset, where its performance approaches that of PSSM. To ensure the reliability of CataPro's results on these four DMS datasets, we calculated the maximum sequence identity between the enzyme sequences in these four test sets and those in our $k_{cat}$, $K_m$, and $k_{cat}/K_m$ datasets. With the exception of TtIGPS, which has a maximum sequence identity of 0.48, the other three enzymes are included in our dataset. Since our ten-fold cross-validation datasets are clustered based on a protein sequence similarity threshold of 0.4, each enzyme and its similar enzyme sequences are only present in the same subset. Supplementary Table 4 presents the performance of all CataPro sub-models for kcat, Km, and kcat/Km on these four DMS datasets. In each DMS dataset, the performance of models trained on subsets where the test enzyme is not included is highlighted in bold in Supplementary Table 4. Interestingly, models that are unfamiliar with the test enzyme are not necessarily worse than those models where the test enzyme is included in the training set. Notably, in the EcTL dataset, the model that had not

encountered the EcTL enzyme achieved a SCC of 0.437, ranking second among all the sub-models. These results highlight the robustness of CataPro for protein fitness landscape prediction.

## CataPro-assisted enzyme mining and directed evolution

In previous assessments, CataPro exhibited notable superiority over the other models. To determine how it performs in real-world applications, we applied CataPro to discovering enzymes and identifying high activity mutations for vanillin biosynthesis.

Vanillin is an important aromatic aldehyde with a rich milky and vanilla aroma, widely used in industries such as food, beverages, cosmetics, and pharmaceuticals[74–77]. Although vanillin is found in many plants, such as vanilla bean pods, its natural production is very low because of the stringent cultivation requirements of these plants, resulting in naturally derived vanillin accounting for less than 1% of the vanillin sold on the market. Over 99% of the vanillin on the market is chemically synthesized using eugenol and guaiacol as substrates[77]. This method is cost-effective but involves high energy consumption and substantial pollution. In contrast, producing vanillin through biotransformation processes using agricultural residues, lignin, and ferulic acid as substrates complies with food safety requirements set by the United States and the European Union, which stipulates that vanillin synthesized from ferulic acid can be considered natural-equivalent vanillin. A synthetic pathway with strong industrial application potential involves the decarboxylation of ferulic acid by ferulic acid decarboxylase (FDC) to produce 4-VG, followed by oxidation by *Caulobacter segnis* carotenoid cleavage oxygenases (CSO2)[75,78]. Here, we demonstrate the high potential of CataPro to assist in the enzyme discovery for highly active oxygenases and mutants for vanillin biosythesis.

To discover more potent 4-VG oxidation enzymes, we employed BLAST to retrieve 1500 sequences with sequence similarity higher than 0.2 to the CSO2 sequence from the UniProt database[60]. Subsequently, sequences with a length difference higher than 20 amino acids compared to the CSO2 sequence were removed. The remaining sequences were screened based on the $K_m$ and $k_{cat}$ values predicted by CataPro, and the top 150 enzymes were retained. As protein function depends on its structure, and structural-based clustering is a useful strategy for enzyme discovery[19], we calculated the structural similarity between these enzymes and CSO2 using TM-Align[79]. The top half of the enzymes were clustered through t-SNE to produce the final five representative enzymes (Fig. 7a). These five enzymes, PpCSO, MgpCSO, PgCSO, SsCSO, and TkCSO, are derived from *Pseudomonas putida*, *marine gamma proteobacterium*, *Pseudomonas gingeri*, *Sphingobium sp.*, and *Trebonia kvetii*, respectively. Among them, only PpCSO is included in

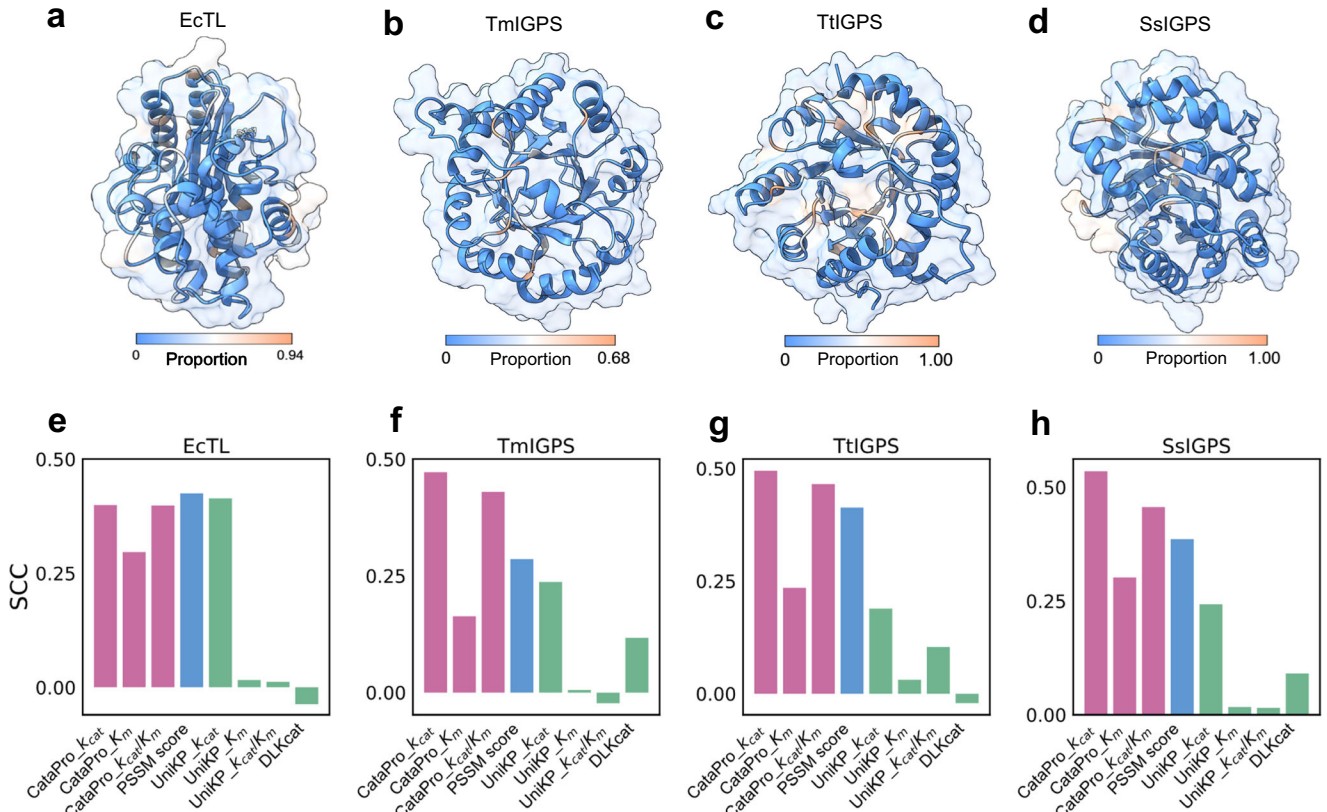

**Fig. 6 | Performance of CataPro and baseline models on the DMS datasets. a-d** respectively show the 3D structures of the EcTL, TmIGPS, TtIGPS, and SsIGPS enzymes, as generated by AlphaFold2. Residues are colored based on the proportion of advantageous mutants at each site (blue-white-red). A redder color indicates a higher proportion of advantageous mutants, while a bluer color indicates a lower proportion. Residues lacking fitness data are also marked in blue. **e-h** respectively show the SCCs achieved by predictors on the EcTL, TmIGPS, TtIGPS, and SsIGPS datasets. In fact, $K_m$ values and PSSM scores are inversely relative to activity. For clarity in comparison, we depict the negative values of the original SCC achieved by $K_m$ values from CataPro and UniKP, as well as PSSM scores, in the bar chart. UniKP and DLKcat correspond to the versions reported in the original paper. Source data are provided as a Source Data file.

the CataPro dataset, while the maximum sequence identity between the other four enzymes and those in our dataset is below 44%. We measured the activity of these five enzymes through experiments and found that the activity of SsCSO was 19.53 times higher than the reference enzyme CSO2 (Fig. 7c).

To validate the efficacy of CataPro in enzyme engineering, we utilized CataPro for computationally driven directed evolution of SsCSO. We applied CataPro and PSSM scores to select mutants, ensuring they simultaneously exhibit high activity and evolutionary conservation simultaneously[80–82]. In practical enzyme engineering, maintaining or enhancing the structural stability of enzymes is equally important[83,84]. To ensure that mutations do not compromise enzyme stability, PSSM serves as a crucial criterion adopted in our approach to restrict the potential protein structure change and reduce the vast mutation sampling space. This strategy has been successfully applied in many enzyme activity and selectivity improvement scenarios[81,82,85]. Residues situated within the binding pocket and directly interacting with the substrate are often targets for mutation, as they directly affect the enzyme function and activity[83,86]. Here, we first employed AlphaFold2[22] to predict the SsCSO structure, followed by molecular docking[87] to simulate the enzyme-substrate complex structure. Ninety-one residues within a distance of 12 Å from the substrate (excluding histidines in contact with the iron ion) were selected as mutation sites. Each site was mutated to the other 19 standard amino acids, resulting in a total of 1729 (19 × 91 = 1729) single-point mutants. The top half of the mutants, ranked by the predicted $k_{cat}/K_m$ values from CataPro, were

retained. Subsequently, we assessed the evolutionary conservation of these mutants using PSSM. If the amino acid at site $n$ mutates from $i$ to $j$, the change in PSSM score caused by this mutation is defined as $\Delta PSSM_{ij}^n = PSSM_j^n - PSSM_i^n$. The larger $\Delta PSSM_{ij}^n$ indicates that the mutation is more consistent with evolutionary trends. Six mutants with $\Delta PSSM_{ij}^n$ greater than 7 were selected for experimental validation (Fig. 7b). Among these mutants, T216M and M351F exhibit higher activity compared to the wild-type (Fig. 7c).

To further enhance the activity of SsCSO, we conducted the next round of mutations based on the combined mutant T216M-M351F. We selected 22 residues in the loop region of the SsCSO pocket (excluding residues duplicated in the first round of mutations) as new mutation sites, resulting in a total of 418 (19 × 22 = 418) mutants. By combining the predicted $k_{cat}/K_m$ values from CataPro and the PSSM scores, six mutants with T216M-M351F as the template were selected for experimental validation (Fig. 7b). The results demonstrated that the mutants T216M-M351F-Q100G and T216M-M351F-V384G exhibited significantly higher activity, being 3.16-fold and 3.34-fold higher than that of the wild-type SsCSO, respectively. When compared to CSO2, the activity improvement is 61.71-fold and 65.23-fold for SsCSO T216M-M351F-Q100G and T216M-M351F-V384G, respectively (Fig. 7c). The relative activity values of all candidate enzymes discovered during enzyme mining and SsCSO mutants generated through enzyme engineering are presented in Supplementary Table 5. Figure 7d illustrates the locations of the final dominant mutation sites. The above experiments demonstrated that CataPro is an effective method for enzyme discovery and engineering.

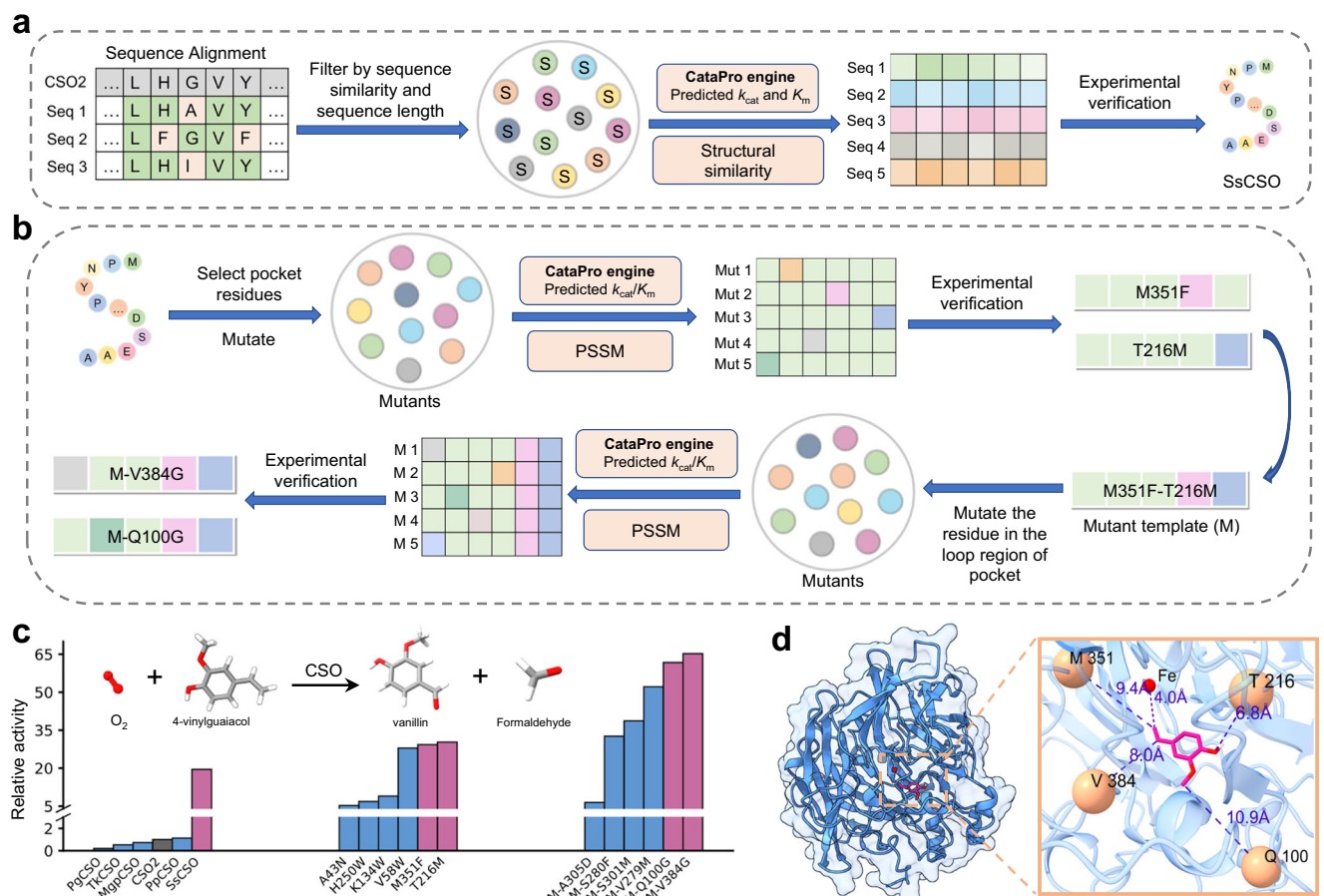

**Fig. 7 | Results of CataPro in the mining and engineering of Carotenoid Cleavage Oxygenases. a** and **b** illustrate the workflows for enzyme discovery and engineering, respectively. **c** The bar group on the left shows the relative activity of the mined candidate enzymes compared to CSO2. The middle and right bar groups respectively show the relative activity of SsCSO mutants compared to CSO2 in two rounds of enzyme engineering. The top subplot illustrates the reaction of CSO

enzymes catalyzing 4-VG to produce vanillin. All activity values are relative to CSO2, with CSO2 activity set to 1. Source data are provided as a Source Data file.
**d** visualizes the locations of several experimentally validated beneficial mutation sites in the SsCSO. Orange spheres denote the mutated residues, the red sphere represents the iron atom, and the pink stick represents 4-VG.

## Discussion

In industrial production involving enzymes, the catalytic efficiency of enzymes has a direct impact on the production cycle and cost of projects. Therefore, the discovery of native enzymes as well as their mutants with high activity for specific biochemical reactions in a timely manner is a key research interest for synthetic biology companies and institutes. With the recent development of computational techniques, especially the iterative improvement of DL algorithms and their remarkable potential in numerous downstream tasks, DL is increasingly being used to assist in enzyme mining and evolution in the field of synthetic biology. Recently, several enzyme kinetic parameter prediction models have been developed, such as DLKcat[41], UniKP[43], DLTKcat[88], and MPEK[44]. Although these models have demonstrated SOTA performance on some datasets, unfortunately, they have overlooked the impact of information leakage from the training set; consequently, their "good performance" is actually merely the result of information leakage between the datasets. Recently, a paper reported that DLKcat demonstrates good accuracy only for enzymes that are highly similar to the proteins used for training, while it exhibits poor performance for mutants and unfamiliar proteins[66]. In reality, enzymes with annotated kinetic parameter data in current open-source databases represent just a small fraction of the vast enzyme reaction space in nature. Similarly, the enzyme sequence space in current enzyme databases is also negligible compared to the entire genome space. In practical applications, the enzyme under consideration is usually less

well-researched. Therefore, the goal of enzyme kinetic parameter prediction models should be to generalize to enzyme-catalyzed reactions beyond the training set, rather than overfit to samples in existing databases. To address this lacuna, we developed ten-fold unbiased datasets for $k_{cat}$, $K_m$, and $k_{cat}/K_m$ based on an enzyme sequence similarity threshold of 0.4. The ten-fold cross-validation was performed on these unbiased datasets to evaluate the real performance of the models. Based on our results section, DLKcat[41] and UniKP[43] both exhibited good performance in the randomly partitioned ten-fold cross-validation datasets, but demonstrated less accuracy in the unbiased ten-fold cross-validation datasets. This indicates that the earlier model is more prone to overfitting. Additionally, it is important to note that the results in Fig. 2 represent the models' performance on all validation sets during training on the ten-fold cross-validation dataset. Since the validation sets were used in hyperparameter optimization during training, the evaluation of CataPro may be slightly overoptimistic, and it may still encounter challenges when applied to highly unfamiliar proteins, reactions, or other independent test sets.

Another key point in building kinetic parameter prediction models is the representation of enzyme-catalyzed reactions. One main approach is to convert information about enzymes and substrates into feature vectors, and then input the resulting feature vectors into ML or DL models to fit the labels. Li et al.[41] split a protein sequence into an overlapping n-gram of amino acids and then translated protein sequences into various word embeddings. A convolutional neural

network (CNN)[89] is employed to extract information from enzymes, while a graph neural network (GNN)[90] is utilized to learn the topology of substrates. DLTkcat adopts a similar feature modeling approach and network architecture[88]. Another method involves utilizing the embedding of protein language models as enzyme features, and the embedding of compound language models as substrate features. For example, UniKP employs the ProtT5-XL-UniRef50 model to generate enzyme representations and the SMILES transformer model to generate substrate representations[43]. MPEK[44] attempted several combinations of embeddings extracted from protein language models and molecular language models, but unfortunately, they still overlooked model overfitting caused by training data leakage. Another study, recently reported, has utilized transfer learning and ensemble learning to improve the performance of enzyme kinetics parameter prediction models[91]. In this work, we systematically tested the performance of embeddings from various language models and molecular fingerprints for predicting enzyme kinetic parameters using the unbiased datasets we developed. The results indicate that embeddings of ProtT5-XL-UniRef50[57] exhibit superior performance compared to those from ESM2[63], particularly in $k_{cat}$ prediction. We visualized the latent space of samples processed by the $k_{cat}$ model using principal component analysis and observed distinct stratification between high and low $k_{cat}$ values (Supplementary Fig. 13). This suggests that the embedding captures the approximate range of sample $k_{cat}$ values. As is well-known, the 3D structure of proteins determines their function. Therefore, we attempted to verify the impact of structural information on predicting enzyme kinetic parameters. We used embeddings from a recently reported protein language modelling-based method, SaProt[55], trained on AlphaFold2 structures, as structural features integrated into enzyme reaction information. However, this did not yield the expected results, possibly because current structure pre-trained protein language models have not yet reached the stage of practical utility.

An important task in enzyme engineering is to modify wild-type enzymes, which requires the model to identify high-activity mutants among numerous variants[92–94]. Previous studies have primarily focused on reporting model performance on datasets encompassing all enzyme-substrate reactions, but have not proposed a benchmark for evaluating the capability of models in enzyme engineering. In this study, we propose two evaluation criteria. The first is the SCC achieved by the model across all mutations within a reaction, which assesses the capability of models to rank mutations based on kinetic parameters. The second criterion is the accuracy in identifying the better-performing mutant between any two mutants within a reaction. Figure 4 demonstrates that CataPro has higher potential in enzyme engineering compared to UniKP[43] and DLKcat[41]. The results in Fig. 5b–d show that CataPro maintains excellent performance on small datasets collected from the literature, demonstrating its generalization ability. We encourage future research on predicting enzyme kinetic parameters to use the dataset and evaluation criteria we have established to assess the ability of models in enzyme engineering. However, it is important to note that the current CataPro faces limitations in accurately capturing the absolute values of mutation effects, despite achieving high correlations in certain reactions. This observation may indicate that accurately predicting enzyme mutation effects remains a considerable challenge.

Crucially, with the assistance of CataPro, we discovered an enzyme called SsCSO, which exhibits an activity 19.53 times higher than that of CSO2 in catalyzing the conversion of 4-VG to vanillin as well as formaldehyde. For modifying SsCSO, we applied CataPro and PSSM to identify a highly active mutant, T216M-M351F-V384G, which exhibits 3.34 times the activity of the wild-type SsCSO. These results indicate that CataPro holds considerable promise for practical enzyme discovery and enzyme engineering processes.

Despite CataPro demonstrating high application value in numerous test scenarios and practical enzyme engineering projects,

it is important to acknowledge that it has certain limitations. For instance, in $k_{cat}$ prediction, the accuracy of CataPro still has some room for improvement. This is mainly owing to two factors: first, the existing $k_{cat}$ data do not adequately cover a wide range of enzymatic reactions; second, current methods for modeling enzyme-substrate reactions encounter challenges in accurately quantifying enzyme catalytic processes and mechanisms. Actually, the catalytic mechanisms of enzymes are not singular. In some cases, active sites utilize the charge accumulation in the transition state to enhance electrostatic interactions, while others take advantage of charge dispersal to enhance the stability of the transition state within a relatively nonpolar pocket[95]. In addition, some utilize general acid-base catalysis, while others use metal ions[96]. Encoding these physicochemical mechanisms may help further improve the performance of ML models. Enzymes are dynamic entities, whose dynamic properties are clearly linked to their biological functions. However, currently, reproducing kinetic information solely based on existing data is a challenge. Developing reliable protein structure pre-trained models capable of effectively capturing protein properties may help alleviate this issue. It is gratifying to note substantial progress in DL in the field of complex modeling. Using DL to model enzyme-substrate binding patterns may help in advantageous in mitigating some of the aforementioned challenges. Another limitation is that CataPro predicts enzyme kinetic parameters for an enzyme and a single substrate, whereas many reactions involve multiple substrates. This means that CataPro relies on partial reaction information rather than complete reaction information, which may introduce the risk of systematic errors due to missing data. Therefore, collecting more data points that include comprehensive reaction information to train a model based on full reaction details remains a very promising direction.

## Methods
### Dataset preparation
We downloaded all entries containing $k_{cat}$ or $K_m$ values from BRENDA (release 2023.1) and SABIO-RK (database v.2.20), which include enzyme commission (EC) numbers, types, organism information, substrate names, UniProt IDs, and $k_{cat}$ and $K_m$ values. The amino acid sequences of enzymes were downloaded from the UniProt database[60] using their UniProt IDs. If an enzyme had multiple UniProt IDs, the data point was discarded. The canonical SMILES string of the substrate was downloaded from PubChem[61] on the substrate names. Among all the kcat and Km entries collected from the BRENDA and SABIO-RK databases, a total of 30,691 unique substrate names were included, but 17,306 of these could not be assigned SMILES. After downloading all the necessary data, we conducted several rounds of data cleaning. First, entries with ambiguous units were removed. Furthermore, for enzyme sequence-SMILES pairs with multiple $k_{cat}$ and $K_m$ annotations, only the maximum $k_{cat}$ value and the minimum $K_m$ value were retained. Ultimately, our dataset contains 27,658 unique $k_{cat}$ entries, and 42,018 unique $K_m$ entries, with 25,831 samples containing $k_{cat}$ as well as $K_m$ values. In the final $k_{cat}$ and $K_m$ datasets, there are 3490 $k_{cat}$ and 6468 $K_m$ data points exhibiting multiple labels from the original BRENDA and SABIO-RK databases. To assess the label noise for these data points, we computed the standard deviation (std) of all original labels for each data point as a measure of label noise. The noise distribution of multi-label data points in the $k_{cat}$ and $K_m$ datasets is shown in Supplementary Fig. 3g-h. The reactions that are present in both the $k_{cat}$ and $K_m$ datasets constitute the $k_{cat}/K_m$ dataset.

To create unbiased datasets for $k_{cat}$, $K_m$, and $k_{cat}/K_m$ prediction tasks, we employed the CD-HIT program[59] to cluster each category of data based on an enzyme sequence similarity of 0.4, then divided all clusters into ten subsets. Thus, unbiased ten-fold cross-validation datasets for $k_{cat}$, $K_m$, and $k_{cat}/K_m$ prediction were prepared.

## Encoding for enzymes and substrates

The encoding of enzyme and substrate information, or enzyme-catalyzed reactions, is a key challenge in predicting enzyme kinetic parameters. In this study, the enzyme information is represented by embeddings of 1024 dimentions generated by ProtT5-XL-UniRef50[57]. For the structure information of substrates, we employed two features, namely embeddings from MolT5 (molt5-base-smiles2caption)[62] and MACCS keys fingerprints, with dimensions of 768 and 167, respectively. We also attempted to use other methods for enzyme and substrate representation. For instance, we utilized embeddings from the ESM2 (esm2_t33_650M_UR50D) pre-trained model to represent enzymes[63], and embeddings from the Mole-BERT pre-trained model[64], Morgan fingerprints, and RDKit fingerprints to represent substrates. The MACCS keys, Morgan, and RDKit fingerprints were all generated using the RDKit package[97].

## Construction of CataPro models

For the $k_{cat}$ and $K_m$ prediction tasks, CataPro consists of two parts: the feature extraction module and the prediction module. For features such as embeddings from pre-trained models and molecular fingerprints, we found that excessive network complexity or a large number of parameters can easily lead to overfitting. Therefore, after extracting features from enzymes and substrates, only one hidden layer and one output layer are used to fit the relationship between the features and $k_{cat}$ or $K_m$. RMSE is used as the loss function.

The training of the $k_{cat}/K_m$ model is conducted in two steps. First, we pre-train the $k_{cat}$ and $K_m$ models on the ten-fold cross-validation dataset for $k_{cat}/K_m$. After training, their weights are fixed. Then, we obtain the initial $k_{cat}/K_m$ values using the pre-trained $k_{cat}$ and $K_m$ models from the previous step. Subsequently, a decoding module based on attention mechanism is utilized to learn a correction term as a supplement to the initial $k_{cat}/K_m$. The attention mechanism is shown in Eq. 1-2, where $h$ represents the input features, $W$ is the weight matrix of the attention layer, $A$ denotes the normalized attention weights, and $\odot$ represents element-wise multiplication. The final $k_{cat}/K_m$ value is obtained by combining the initial $k_{cat}/K_m$ value with the correction term using certain weights.

$$\mathbf{A} = Softmax(\mathbf{W} \cdot \mathbf{h}) \tag{1}$$

$$\mathbf{h}_A = \mathbf{A} \odot \mathbf{h} \tag{2}$$

We employed grid search to identify the optimal parameters for the CataPro, UniKP, and DLKcat models. The hyperparameter search spaces and the final selected parameters for these models during training are presented in Supplementary Tables 6–8.

## Sequence identity calculation

The sequence identity between the target enzyme and enzymes in our dataset, as described in Sections 2.5, 2.6 and 2.7, was calculated using the blastp tool in BLAST.

## Evaluation metrics

In this work, three evaluation metrics were primarily utilized, namely Pearson correlation coefficient (PCC), Spearman correlation coefficient (SCC), and root mean square error (RMSE):

$$PCC = \frac{\sum_{i=1}^{n}(X_i - \bar{X})(Y_i - \bar{Y})}{\sqrt{\sum_{i=1}^{n}(X_i - \bar{X})^2}\sqrt{\sum_{i=1}^{n}(Y_i - \bar{Y})^2}} \tag{3}$$

$$SCC = 1 - \frac{6\sum d_i^2}{n(n^2 - 1)} \tag{4}$$

$$RMSE = \sqrt{\frac{1}{n}\sum_{i=1}^{n}(Y_i - X_i)^2} \tag{5}$$

Here, $X_i$ and $Y_i$ are predicted and experimental $\log_{10}k_{cat}$ or $\log_{10}K_m$, or $\log_{10}(k_{cat}/K_m)$. $\bar{X}$ and $\bar{Y}$ represent the mean predicted value and the mean experimental value, respectively. $d_i$ is the difference in ranking between the predicted and experimental value for a particular sample.

When evaluating the models' ability to identify the better-performing mutant between any two mutants for a given reaction, we first generate the all pairwise combinations (order-independent) of mutants for the reaction. For a reaction with $N$ mutants, the total number of pairwise combinations is $M_{total} = N(N-1)/2$. For each pair, if the model accurately identifies which of the two mutants performs better, the prediction is considered successful. By evaluating all pairwise combinations within the reaction and counting the number of successful predictions ($M_{true}$), the accuracy is defined as follows:

$$Accuracy = \frac{M_{true}}{M_{total}} \tag{6}$$

## Calculating PSSM based on multiple sequence alignment

PSSM is a widely used in bioinformatics for protein sequence alignment. It is derived from multiple sequence alignments and provides a statistical representation of the conservation and variation of amino acids at each position in a protein sequence. The PSSM score at a given position indicates the likelihood of finding a particular amino acid at that position based on evolutionary information from aligned sequences. Higher positive scores indicate a higher degree of conservation, while negative scores suggest variability. PSSM scores are useful for identifying conserved regions and potential functional sites. In this work, we first applied Jackhmmer[98] to search for homologous sequences of SsCSO from UniRef90 (April 2024)[99], generating a multiple sequence alignment, based on which a PSSM score matrix was generated using the PSSM calculation tool in TGT_Package [https://github.com/realbigws/TGT_Package.git][100].

## Visualization of AlphaFold2 structures

The structures of the four enzymes in the DMS datasets (EcTL, TmIGPS, TtIGPS, SsIGPS), as well as the SsCSO structure we identified, are all sourced from the AlphaFold database[22,101]. The predicted local-distance difference test (pLDDT) and predicted aligned error (PAE) for these five AlphaFold2 structures are shown in Supplementary Figs. 14–15. The UniProt IDs and DOIs in the AlphaFold database for these five proteins are as follows: (P62593, https://www.alphafold.ebi.ac.uk/entry/P62593), (Q56319, https://www.alphafold.ebi.ac.uk/entry/Q56319), (P84126, https://www.alphafold.ebi.ac.uk/entry/P84126), (Q06121, https://www.alphafold.ebi.ac.uk/entry/Q06121), and (A0A243P926, https://www.alphafold.ebi.ac.uk/entry/A0A243P926).

## Experimental materials and chemicals

The plasmid pET28a(+) used in the experiment was purchased from Tongyong Co., Ltd. (Shanghai, China), and the *Escherichia coli* BL21(DE3) competent cells were obtained from WeiDi Chemical Co. Ltd. (China). The strains involved in the experiment were constructed in our laboratory. The chemicals used in the experiment are listed in Supplementary Table 9.

## Gene Cloning and Expression

From the protein sequences, we derived the DNA sequences of CSO genes through codon optimization using DNAworks[102]. The amino acid sequence and DNA sequence of CSO enzyme are shown in Supplementary Data 1 and Supplementary Data 2, respectively. Subsequently, the synthesized CSO gene sequences were inserted into the NdeI and

XhoI sites of the pET28a(+) vector following the guidelines provided in the Hieff Clone® Plus One Step Cloning Kit (Cat#10911) from Yisheng Biotechnology Co., Ltd. (Shanghai, China). The resultant recombinant expression vector was then introduced into E. coli BL21(DE3). A single colony was picked using a sterile pipette tip and resuspended in 20-50 $\mu$L of LB medium. Then, 1 $\mu$L of the suspension was directly used as the PCR template. Colony PCR was performed using the universal primers T7 (5′-TAATACGACTCACTATAGGG-3′) and T7t (5′-GCTAGTTATTGCT-CAGCGG-3′) to identify positive clones, following the instructions of the 2 × Hieff® HotStart PCR Genotyping Master Mix (With Dye) (Cat. No. 10108ES50) provided by Shanghai Yeasen Biotechnology Co., Ltd. The positive clones were then sent for third-party sequencing, performed by Shanghai General Biosystems Co., Ltd., using T7 and T7t as sequencing primers. Clones with accurate DNA sequences were selected. A single clone was selected and cultured in 4 mL of liquid LB medium supplemented with 50 $\mu$g/mL kanamycin, incubating at 37 °C with agitation at 220 rpm for 6 hours. The culture was then transferred at a 2% inoculation ratio into 50 mL of liquid 2YT medium containing 50 $\mu$g/mL kanamycin and further cultured at 37 °C with agitation at 220 rpm until the OD600 reached approximately 1.0. IPTG was introduced to achieve a final concentration of 0.2 mM, along with $FeCl_2$ to reach a final concentration of 1 mM, and induction was carried out at 25 °C with agitation at 200 rpm for 16 hours. Cells were harvested through centrifugation at 4 °C and 12,000 × g for 5 minutes. The induced cells were resuspended in a buffer solution (50 mM pH 7.0 PB, 150 mM NaCl, 10% glycerol) and subjected to sonication (50% power, 2 seconds on, 3 seconds off, totaling 15 minutes of sonication). Following this, the cell debris was eliminated via centrifugation at 4 °C and 12,000 × g for 20 minutes, and the resulting supernatant was collected as the crude enzyme extract for assessment of soluble expression levels through SDS-PAGE analysis.

### Protein purification and validation
The CSO crude enzyme extracts were purified using nickel ion affinity chromatography (Ni-NTA resin, GE Healthcare, Fairfield, San Diego, USA) and the AKTA Prime Plus system (GE Healthcare, Fairfield, San Diego, USA) with a 50−400 mM imidazole gradient at a flow rate of 1.0 mL/min. The SDS-PAGE gel images are provided in Supplementary Fig. 16.

### Determination of enzymatic activity
The enzyme activity assay was conducted in a 200-$\mu$L reaction system, consisting of 60 mM 4-VG, 10 $\mu$L of ten-fold diluted crude enzyme solution, and 50 mM pH 9.5 $Na_2CO_3$-$NaHCO_3$ buffer. The reaction mixture was incubated at 25 °C for 60 minutes, after which 20 $\mu$L of 2.5 M HCl was added to terminate the reaction. The mixture was then centrifuged at 1950 × g for ten minutes. In a 96-well microplate, 50 $\mu$L of 2,4-dinitrophenylhydrazine (8 g/L, dissolved in 2.5 M HCl) was added to 10 $\mu$L of the supernatant from the reaction mixture. The solution was shaken at room temperature for 3 minutes, followed by the addition of 80 $\mu$L of 4 M NaOH solution, and then shaken again at room temperature for 2 minutes. The absorbance at 480 nm was then measured to evaluate the enzyme relative activity.

### Construction of mutant plasmids
Upstream and downstream primers were designed according to the mutation sites. The primer sequences are listed in Supplementary Table 10. Using the plasmid pET28a(+)-CSO2 or respective wild-type or mutant CSO genes as the templates, amplification was performed with GXL DNA polymerase (TaKaRa, Dalian, China) under the following conditions: initial denaturation at 94 ° for 2 minutes, 98 ° for 10 seconds, annealing at 55 ° for 15 seconds, and extension at 68 °C for 6 minutes and 20 seconds. After 25 cycles, a final extension was performed at 68 ° for 10 minutes. PCR products were analyzed by agarose gel electrophoresis. The resulting products were then digested with

the restriction enzyme DpnI (Thermo Scientific) to digest the plasmid at 37 ° for 30 minutes, followed by denaturation at 80 ° for 5 minutes. The digested products were purified using a DNA purification kit. Cloning was performed as per the instructions of the Hieff Clone® Plus One Step Cloning Kit (Cat#10911) to obtain the recombinant expression vector, which was then transformed into E. coli BL21(DE3). Colony PCR and DNA sequencing (by Tongyong Co., Ltd., Shanghai, China) was performed to screen for positive clones. The positive clones were further cultured to obtain crude enzyme solutions of the successful mutants, and enzyme activity was measured.

### Reporting summary
Further information on research design is available in the Nature Portfolio Reporting Summary linked to this article.

## Data availability
The information for the ten-fold cross-validation datasets that we created is sourced from BRENDA[32], SABIO-RK [http://sabio.h-its.org/][33], UniProt [https://www.uniprot.org/][60], and PubChem [https://pubchem.ncbi.nlm.nih.gov/][61]. The TAL homologue dataset and TAL engineering dataset are collected from https://doi.org/10.1038/s41467-023-44113-1[43] and 10.1093/bib/bbae387[44]. The DERA dataset and BH1352 dataset are sourced from 10.1126/science.1063601[68] and 10.1074/jbc.RA119.011363[69], respectively. The mutation dataset of EcTL is from 10.1093/molbev/msu081[71], and the mutation datasets of TmIGPS, TtIGPS, and SsIGPS are all from 10.1038/ncomms14614[72]. The unbiased ten-fold cross-validation datasets we created are publicly available in the Zenodo repository at [https://zenodo.org/records/14894710] and the GitHub repository at [https://github.com/zchwang/CataPro]. Source data are provided with this paper. Unless otherwise stated, all data supporting the results of this study can be found in the article, supplementary, and source data files. Source data are provided with this paper.

## Code availability
The source code and detailed documentation for CataPro are stored at [https://github.com/zchwang/CataPro] and are publicly available.

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

## Acknowledgements

This research is supported by National Key R&D Program of China (2023YFA0915500 to S.W. and L.Z.). The project is also supported by Taishan Scholars Program of Shandong Province (tsqn202312159 to Y.Y.), Taishan Scholars Program of Shandong Province (tstp20240807 to W.L.), Shandong Provincial Natural Science Foundation under Grant (No. ZR2024MA071 to W.L.), Shenzhen Science and Technology Program (KJZD20230923115901003 to X.L.), and SIAT Distinguished Young Scholars(E4G021 to X.L.).

## Author contributions

L.Z. and W.L. designed and supervised the project. Z.W. designed CataPro and performed the computational experiments. D.X. performed the wet-lab experiments. Z.W., D.W., X.L., S.W., Y.L., Y.Y., W.L., and L.Z. analyzed data. Z.W., D.X., and L.Z. wrote the manuscript. Z.W., Y.Y., W.L., and L.Z. revised the manuscript.

## Competing interests

The authors declare that no competing interests.
