## [Transparent Peer Review file · Nature Communications]

Robust Enzyme Discovery and Engineering with Deep Learning using CataPro

Corresponding Author: Liangzhen Zheng

Version 0:

Reviewer comments:

Reviewer #1

(Remarks to the Author)

I provide my review in the attached .docx file.

(Remarks on code availability)

If I see it correctly, the training and validation datasets are not provided. Without these the study is not reproducible.

Reviewer #2

(Remarks to the Author)

This article presents a novel approach by integrating enzyme kinetics and transfer learning, offering a new tool for enzyme discovery and directed evolution. Instead of using the traditional ratio of K_{cat} to K_m to predict catalytic efficiency (K_{cat}/K_m), the authors directly predict K_{cat}/K_m through a transfer learning model. This innovative method not only improves the accuracy of predictions but also simplifies the process, demonstrating significant potential for practical applications. The study validates the model's accuracy and utility through the discovery and modification of CSO enzymes, providing a fresh perspective and methodology for the field of enzyme engineering. Compared to existing kinetic models, the proposed transfer learning approach opens new avenues for handling large enzyme databases, with the potential to significantly impact the efficient discovery of novel enzymes and optimization of enzyme activity. Despite the many innovations in this article, there are some minor questions.

1. The first paragraph of the introduction only discusses the time and cost associated with obtaining high-activity enzyme variants, without adequately addressing the challenges in enzyme discovery. However, considering the discussion in section 2.7 of the results, it is recommended to include a discussion of the challenges commonly encountered in enzyme discovery within the introduction, as well as how artificial intelligence technology can significantly advance enzyme mining efforts. This would better align with the paper's focus on "enzyme discovery" as reflected in the title and provide a more comprehensive background for the readers.
2. The transition between the second and third paragraphs of the introduction is weak, making the structure feel disjointed. It is suggested to introduce transition sentences between these paragraphs to enhance the logical connection and create a smoother flow, allowing readers to more easily follow the background and significance of the research.
3. Please ensure consistent decimal places in the figures. For example, in Figures 2 and 3, the number of decimal places varies.
4. The data set in Fig. 3a shows that K_{cat}/K_m values exhibit a normal distribution. This raises two concerns: first, if a given enzyme's K_{cat}/K_m falls outside this distribution, can the model's predictions still maintain accuracy? Second, based on this distribution, does this imply that the prediction is only accurate when the results fall within this range?
5. Result 2.7 Apart from CSO enzymes, could the mining and modification of other more common enzymes be discussed to demonstrate the accuracy of the model? Additionally, are there sequences in the training set that are similar to the CSO enzymes?
6. Line 304. In the CSO mining process, sequences with a length difference higher than 20 amino acids from CSO2 were excluded. What is the basis for this deletion? This approach may have led to the loss of potentially higher-activity sequences.
7. Fig. 5. The authors compare the predicted values of the mutant enzymes with the experimental values of the wild-type

enzyme to demonstrate the model's accuracy. However, I believe this approach may introduce bias. The predicted activity of the wild-type enzyme should be used as the reference instead of the experimental values. If only predicted values are used for comparison, will the results still adhere to expected trends?

(Remarks on code availability)

Reviewer #3

(Remarks to the Author)

The manuscript "Robust Enzyme Discovery and Engineering with Deep Learning using CataPro" by Wang et al. presents a fair and unbiased dataset to evaluate models predicting enzymatic kinetic parameters, as well as a framework for predicting these parameters. The question addressed is important and likely of interest to a broad audience. The authors conducted many experiments (comparison of different methods, architecture choices...) with several datasets (k_cat, Km, kcat/Km, DMS...) in different conditions (biased, unbiased...). The code is available for all in open access. However, I do have major concerns regarding the original contribution with respect to previous work and the comparison with SOTA.

- The position with respect to UniKP (ref 17, at least one author in common with the present manuscript) is not clearly stated. Specifically the authors claim that "Unlike previous studies, we also conducted the first evaluation of the capability of the enzyme kinetic parameter prediction model to identify mutation effects." but this does not seem true since mutation effects were already predicted in Ref 17. The paragraph and results about "extra trees" are not clear. The authors show that performance of CataPro improves with extra trees. I guess they replaced the linear layer with extra trees, is it the case? The authors' claim that they are using extra trees "instead of neural networks" is very confusing. Then, the authors show that CataPro outperforms UniKP(*) on Fig. 2 but that CataPro (extra trees) has performance similar to UniKP on Supp Fig. 8 (for instance). What has changed between these two experiments? Does the change come from the dataset? Or did the authors initially replace the extra tree in UniKP with a linear? And if the performance are similar, the contribution of CataPro to the state of the art for predicting kcat and Km is questionable.

- More generally, the manuscript lacks specific references to support sentences starting with "Researchers rely on co-evolution...", "some researchers employed...".

- The methods lacks information. How many layers? trainable parameters? What is precisely the architecture of the "decoding module based on attention mechanism"? Is it transformer-based attention?

- What is the level of noise in the experimental data? The authors takes the max value in case there are multiple, but they do not give indication about the extent to which these multiple values are contradictory or consistent. See for instance these study: <https://doi.org/10.1021/acs.jcim.4c00049> on thermodynamic parameters noise across datasets.

- The results on the DMS seem very poor. The SCC values are low compared to what can be seen on ProteinGym leaderboard: <https://proteingym.org/benchmarks>. Particularly, the beta-lactamase is the classical example where variant effect predictors reach very high Spearman Correlation coefficient.

(Remarks on code availability)

Version 1:

Reviewer comments:

Reviewer #1

(Remarks to the Author)

I have attached my review as a docx file to make my responses to the authors' comments more readable.

(Remarks on code availability)

Reviewer #2

(Remarks to the Author)

The previous questions have been answered and I think the manuscript is acceptable.

(Remarks on code availability)

Reviewer #3

(Remarks to the Author)

The authors have successfully addressed my comments.

(Remarks on code availability)

Version 2:

Reviewer comments:

Reviewer #1

(Remarks to the Author)

The authors have successfully and comprehensively addressed all my comments and concerns.

(Remarks on code availability)

Major concerns

1. **In several places the authors discuss the limitations of DLKcat (e.g. ll. 66ff., 80ff., l.351). There is a recent publication that discusses the limitations of DLKcat: <https://doi.org/10.1093/biomethods/bpae061>. This publication should be cited in these places.**

Answer: According to your suggestion, we have added the citation for this publication in the discussion section of the revised manuscript and highlighted it in blue.

2. **Omission of a test set: Using only 10-fold cross-validation without a separate test set can lead to slightly optimistic performance estimates. (i) If you tune your model based on the average cross-validation results, you may inadvertently optimize for those specific folds, leading to overfitting. (ii) Without a separate test set, you never evaluate your model on completely new, unseen data. This doesn't fully mimic real-world scenarios. Using a separate final test set helps address these issues by providing a truly independent evaluation of your model's performance on data it has never seen before. Since generating a separate and independent test set would require redoing the entire analyses, this is obviously not practical or necessary. Moreover, the authors later evaluate the model on additional experimental data, which partially fills the role of independent test data. However, the authors could at least state that the results from the mean cross-validation metrics may be slightly overoptimistic compared to what one would see for truly independent data.**

Answer: Thank you for your reminder and suggestions. We have added the following sentence to the discussion section: “Additionally, although CataPro performs well on the unbiased ten-fold cross-validation dataset, it may still encounter challenges when applied to highly unfamiliar proteins, reactions, or other independent test sets.”, and highlighted it in blue.

3. **When CataPro and UniKP are trained on the same data for kcat and KM prediction, the differences in model performance are small (Figure 2h). This suggests that the main advancement of CataPro for kcat and KM prediction lies in the design of the datasets rather than in a novel model architecture. This should be emphasized in the manuscript.**

Answer: Fig. 2h shows the performance of CataPro (extra tree) and UniKP on both unbiased and biased datasets, where CataPro (extra tree) means replacing linear layer with extra tree in CataPro. Since the only difference between UniKP and CataPro (extra tree) lies in the representation of the substrate, the superiority of CataPro's feature combination can be seen in Fig. 2h, although the advantage is limited, as you mentioned. Furthermore, Fig. 2h provides a more intuitive illustration of the performance differences of the models on biased and unbiased datasets. Here, we urge researchers to focus on the model's performance on unbiased datasets to enable a more objective evaluation. In accordance with your suggestion, we have enhanced the discussion in Section 2.2 in the revised manuscript and highlighted it in blue.

4. **CataPro does not use any reaction information. The turnover number kcat depends on the whole chemical reaction catalyzed by an enzyme and not only on one substrate. However, CataPro only uses information about one of the substrates that is part of the**

reaction. This limitation is not mentioned or discussed anywhere in the manuscript. Why is the entire reaction not encoded? Is this limitation important for the performance of the model? How does one choose a single substrate in the case of reactions with multiple reactions?

Answer: Thank you for your comments. We agree that reaction information is an important factor. We ultimately do not use reaction information, mainly due to the following concerns: (1) Collecting reaction information introduces additional conditions and constraints, which will reduce the sample size of the final dataset. (2) With limited sample size, it will be difficult to support the model in achieving additional functionalities, such as the ability to rank mutants.

When predicting enzyme kinetic parameters for reactions involving multiple substrates, we can select a key substrate as input. In fact, in the BRENDA database, some entries may omit one of the two substrates, and the omitted substrate is typically a common, low-molecular-weight molecule, such as water, oxygen, or metal ions. As discussed in **Section 2.7**, 4-vinylguaiacol is evidently a key substrate in the reaction.

5. kcat/KM prediction architecture:

a. The authors write “this approach [directly predicting kcat/KM] may not achieve optimal performance, as it makes kcat/KM entirely independent of kcat and KM” (l. 130f.). Did the authors test whether their approach actually leads to superior performance compared to a simple model that directly predicts kcat/KM with a single model?

Answer: Yes, we do. In **Fig. 3d-f**, “ProtT5_MolT5-MACC” represents the performance of directly predicting kcat/Km using the architecture shown in **Fig. 1b**. We have added a clearer description of “ProtT5_MolT5-MACC” in **Section 2.3** of the revised manuscript.

b. The authors apply “an attention layer before the output layer [that] extracts crucial information” (l.138). The attention layer is applied to a single vector, right? How is the attention function applied to a single vector? Can the authors give a function?

Answer: Yes. We applied feature-wise attention to a single feature vector to allow the model to focus on the more important features within it. Based on your suggestion, we have added the formula (Eq. 1) in **Section 4.3** to clarify the attention mechanism used here.

c. For the kcat/KM prediction model, models were pre-trained to predict only kcat and only KM. What data were these models trained on? The authors state "we pre-train the kcat and Km models on the tenfold cross-validation dataset for kcat/KM", but the kcat/Km dataset does not contain kcat and KM values. Does this mean that the models are pre-trained on KM and kcat data for enzymes in the kcat/KM training folds for which you can find values in the kcat and KM datasets?

Answer: In our study, the kcat/Km entries are sourced from the kcat dataset and the Km dataset. For instance, if a sample possesses both kcat and Km measurements, it is included in the kcat/Km dataset. Consequently, each entry in the kcat/Km dataset contains experimental kcat and Km values. We have added this description in **Section 2.1** of the

revised manuscript.

- d. **The exact pipeline and architecture of the kcat/KM prediction model is unclear from the main text and methods. It seems that the KM and kcat models are first pre-trained and then fixed, right? Is a model then trained to predict the differences between the predicted kcat/KM (with the above models) and the actual kcat/KM? It should be described in detail when and how which model is trained and how the final prediction is achieved.**

Answer: Thank you for your valuable suggestion. I appreciate your understanding of our models. During the training phase of kcat/Km models, we first pre-trained the kcat and Km models on the kcat/Km ten-fold cross-validation dataset, resulting in ten kcat models and ten Km models. The initial kcat/Km predictions were derived by dividing the predictions from the pre-trained kcat models by those from the pre-trained Km models. However, this initial kcat/Km is not accurate because there are inherent errors in the kcat and Km model. Therefore, we trained a correction term on each fold of the ten-fold cross-validation dataset to compensate for this error. As you mentioned, this correction term is intended to learn the difference between the initial kcat/Km value (calculated by directly dividing the predicted value from the kcat model by the predicted value from the Km model) and the experimental kcat/Km value. When training this correction term, the parameters of the pre-trained kcat and Km models are fixed. According to your suggestion, we have added a detailed description of the kcat/Km model in **Section 2.1** of the revised manuscript.

6. **Comparison with TurNuP and KM prediction model: The authors did not perform a comparison with the TurNuP model, which showed much better performance than DLKcat. While the TurNuP model architecture is not special, the authors put a lot of effort into data cleaning and preprocessing, such as determining the complete chemical reaction equations for all data points. Since TurNuP and CataPro were evaluated on different datasets, a direct fair comparison is not possible. However, if we consider (i) in both cases only test enzymes with a sequence identity of less than 40% compared to the training enzymes and (ii) that the PCC squared is approximately equal to the coefficient of determination R², we obtain the following comparison: CataPro $PCC^2 = (0.497)^2 = 0.247$ vs. TurNuP $R^2 = 0.328$ Performing the same comparison for CataPro and the ProSmith Km (<https://doi.org/10.1371/journal.pcbi.1012100>, see **Minor Concern 2.**) prediction model for test sequences with less than 40% sequence identity, we obtain the following comparison: CataPro $PCC^2 = (0.633)^2 = 0.401$ vs. ProSmith $R^2 = 0.41$ Can the authors comment on why the results (at least in this simplified comparison) do not show significant differences between the two prediction models? TurNuP and ProSmith_KM only provide predictions for wild-type enzymes? Could it be that both models perform similarly for wild-type enzymes?**

Answer: Thank you for your suggestion. To ensure a fair comparison with TurNuP, we trained and tested our model on the dataset used by TurNuP. Since the TurNuP dataset includes all substrates and products for each reaction, we extracted feature vectors for each substrate and product individually. For reactions with multiple substrates or products, we calculated the average of the corresponding feature vectors to serve as the final representation of the substrates

or products. Then, the feature vectors of the enzyme, substrate, and product are concatenated to obtain the representation of the entire reaction (**Supplementary Fig. 9a**). Finally, a linear layer, as used in **Fig. 1b**, is applied to predict *k_{cat}*. We trained CataPro on the five-fold cross-validation dataset divided by Kroll et al., and then evaluated the resulting five models on the four subsets of the test set, which were divided based on the enzyme's maximum sequence identity to those in the training set. The results are shown in **Supplementary Fig. 9b**. An interesting observation is that one of the models (Fold-5) achieved an R^2 of 0.374 in the subset with the maximum sequence identity below 40%, slightly outperforming 0.33 of TurNup. It is important to note that the number of training samples for each sub-model of CataPro is actually four-fifths of the TurNup training samples, as one-fifth of the samples are used for validation. Another notable observation is that, within the subset exhibiting a maximum sequence identity of 99%-100%, the performance of the five CataPro models varies considerably. This discrepancy could be attributed to the fact that each training set is missing one-fifth of the samples from the original dataset. We discussed these results in **Section 2.3** of the revised manuscript.

ProSmith is also an interesting work. The similar performance of ProSmith_KM and CataPro_KM based on your rough comparison might also suggest that the differences among current models for *K_m* prediction are perhaps not significant, as shown in **Fig. 2**, at least smaller than the differences in *k_{cat}* prediction.

- 7. Differences in prediction performance between *K_m* and *k_{cat}*. The authors give the following reasons for the differences in model performance between *K_m* and *k_{cat}* prediction: “These results indicate that predicting *k_{cat}* is more challenging than predicting *K_m*. This is because, in certain enzyme-catalyzed reactions, single-point mutations of enzymes may result in *k_{cat}* values varying by several orders of magnitude. However, mutated enzyme representations often exhibit limited feature differences, especially among similar mutants. Another factor is that the current amount of *k_{cat}* data is small compared to the vast mutation space of enzymes, making it difficult for models to learn the impact of mutations on *k_{cat}*[39].” I find the arguments not very convincing. Mutations can also cause large differences in *K_m* (and differences can also be seen in the wildtype-only models), and I believe the difference in dataset sizes are not big enough to explain the large differences in model performance. My suspicion would be that (i) it is simply a more difficult task to predict *k_{cat}* than *K_m*, and (ii) *k_{cat}* measurements can be more noisy than *K_m* measurements, and hence training data for *k_{cat}* might be more noisy. **Answer:** Thank you for your comments. We agree with your point. Predicting *k_{cat}* is indeed more challenging than predicting *K_m*. *K_{cat}* represents the maximum number of catalytic reactions an enzyme can catalyze per unit time, encompassing the entire catalytic process, including transition state formation, overcoming energy barriers, and product release. Therefore, it is challenging to model this dynamic process exclusively based on enzyme sequences and substrate structures. *K_m* is more like an inherent property of the enzyme towards a substrate, reflecting the enzyme-substrate binding strength to some extent and being generally simpler to model compared to *k_{cat}*. According to your suggestion, we revised this part of the discussion in the revised manuscript.**

8. Analysis in section 2.4:

- a. **Were all mutants and wildtypes analyzed in this section always part of the training set?**

Answer: For each of the ten-fold cross-validation datasets we constructed, each fold contains samples from both the wild-type and its mutants.

- b. **If I understand correctly, the authors evaluated the performance for the mutant kcat prediction for different enzyme families simultaneously. However, this can potentially lead to an overly optimistic PCC as the following simple example demonstrates: Suppose that we have three wildtype enzymes, W1, W2 & W3, with kcat values of 1/s, 100/s, and 1000/s, respectively. We now want to predict the kcat value for three mutants of these wildtypes, M1, M2 & M3, with true kcat values that are significantly different from the corresponding wildtype kcat values: 0.1/s, 130/s and 1300/s, respectively. If we have a model that simply predicts the kcat values of the wild-type enzymes and thus predicts values close to 1/s, 100/s and 1000/s for the mutants, the model clearly cannot predict the effect of the mutations on the kcat value. However, if we compare the raw predicted values with the raw true values for the mutants, we still obtain a very high PCC value due to the large scale of the kcat values, and it appears as though we have an almost perfect mutant prediction model (see figure below). The accuracy of identifying advantageous mutations could also be slightly biased if we assume that most mutants lower the kcat value. A more intuitive and unbiased measure could be to compute for each mutant the difference compared to the corresponding wildtype kcat value and then comparing predicted differences from true differences. Such delta values as computed in <https://doi.org/10.1093/biomethods/bpae061> can then be compared across different enzymes.**

Answer: In fact, what we are evaluating here is the model's ability to rank mutants for each reaction (with the same UniProt ID and SMILES). For example, in **Fig. 4a-c**, we first calculated the SCC achieved by the model for each reaction, and then presented the SCC results across all reactions in the form of a box plot. Therefore, the method you mentioned is actually consistent with the approach used in **Fig. 4a-c** of our manuscript. **Fig. 4d-f** demonstrates the model's ability to predict advantageous and disadvantageous mutants. Taking **Fig. 4a** as an example, advantageous mutants are defined as those where the real kcat value is higher than that of the wild-type. If the model's predicted kcat for this mutant is greater than that for the wild-type, it indicates a successful prediction. We calculated the accuracy of the model for each reaction (with the same UniProt ID-Smiles) and presented the model's performance across all reactions in the form of a box plot. Therefore, these two metrics (SCC and Accuracy) can provide a more intuitive assessment of the model's ability to rank mutants for the specific reaction.

9. **Analyses in Sections 2.5 and 2.6: Did the authors make sure that none of the sequences from the external databases were present in the CataPro dataset to avoid data leakage?**

Answer: In **Section 2.5**, for the two test datasets of TAL, only a five-site mutant of RgTAL is present in our curated dataset, while the sequences of other TAL enzymes have a maximum sequence identity less than 0.67 with the samples in our dataset. For the DERA enzyme, our kcat/Km database includes five reaction data points for its catalysis of other substrates, and the mutation sites involved are all not among those shown in **Fig. 5c**. The last test enzyme, BH1352 (**Fig. 5d**), has a maximum sequence identity of only 0.54 with the enzymes in our kcat/Km dataset.

In **Section 2.6**, three of the four test enzymes (EcTL, TmIGPS, and SsIGPS) are included in our curated datasets, with a maximum sequence identity greater than 0.97, while TtIGPS has a maximum identity of only 0.48 with the enzymes in these datasets. To demonstrate the impact of identical or similar enzyme information in the training set on CataPro, we present the performance of all sub-models obtained from ten-fold cross-validation on these four datasets in the **Supplementary Table 4**. In our unbiased ten-fold cross-validation datasets, identical or similar enzymes are assigned to the same fold in the dataset. Therefore, when these similar enzymes of test sets are included in the validation set, the training set no longer contains enzymes with high similarity to the test enzyme. The performance of models trained on training sets that exclude similar enzymes is highlighted in bold for the corresponding test set in **Supplementary Table 4**. The models corresponding to these bolded values have training sets in which the sequence identity to the test enzyme is less than 0.5. An interesting observation is that several models trained on datasets containing enzymes identical or similar to the test enzyme exhibited poorer performance compared to those trained on datasets that did not include such enzymes. This also suggests that the superior performance of CataPro on these test sets is not due to information leakage. We have added the relevant discussion in **Section 2.6** in the revised manuscript.

- 10. The description of how enzyme candidates were selected during the directed evolution described in Section 2.7 is unclear to me. The authors write “considering the PSSM scores, six representative mutants were selected for experimental validation”, which is vague. Can the authors give specific rules that they followed to select candidates to make the process understandable and reproducible?**

Answer: Thank you for your suggestion. Here, we use the difference in PSSM scores before and after mutation at a given site to assess the tendency of the mutation. If the mutation leads to a significant increase in PSSM score, it suggests that the mutation is more in line with evolutionary trends. Therefore, after screening with CataPro, we selected six mutants with a PSSM score increase greater than 7 for experimental validation. We have provided additional details regarding this process in the revised manuscript.

- 11. An overall comment on the Methods section, which is related to some of my other concerns: The descriptions are very short and partly vague. I think that not enough information is given to successfully reproduce the study.**

Answer: Based on your previous suggestions, we have added more details and explanations in the Results and Methods sections to improve the clarity and understanding for readers.

12. How did the authors handle multimer enzymes with multiple different UniProt IDs? Which of the UniProt IDs was chosen in those cases?

Answer: During data collection, we excluded samples with multiple UniProt IDs.

13. If I see it correctly, the training and validation datasets are not provided. Without these the study is not reproducible.

Answer: We have uploaded our curated datasets to GitHub for free access by everyone.

14. How were hyperparameters of the tested models optimized? Which hyperparameters were tested?

Answer: We used grid search to find the optimal learning rate, batch size, and dropout rate for CataPro. In accordance with your suggestion, we have listed the hyperparameter search spaces and final parameters used for CataPro, UniKP, and DLKcat during training in **Supplementary Tables 6–8**.

Minor concerns

1. Discussion of EITLEM paper: On 19.09.2024 the EITLEM paper was published, a new state-of-the art method for predicting kcat, Km and kcat/KM (full paper: <https://doi.org/10.1016/j.checat.2024.101094>, short preview article: [/www.cell.com/chem-catalysis/fulltext/S2667-1093\(24\)00308-7](http://www.cell.com/chem-catalysis/fulltext/S2667-1093(24)00308-7)). Since the paper was published after the CataPro authors have submitted their manuscript, I think the EITLEM does not need to be discussed or analyzed. However, the authors could add a short paragraph to the discussion, showing their awareness of recent advances and explaining that due to overlaps in the publication timeframes, no comparison was possible. But, as I said I believe this is optional and I just wanted to raise the author's awareness to the similar study.

Answer: Based on your suggestion, we have mentioned this work in the discussion section.

2. The authors refer to a KM prediction (DOI: 10.1371/journal.pbio.3001402), but this was recently improved in another publication (DOI: 10.1371/journal.pcbi.1012100). I hence suggest, the authors cite the new, improved model for predicting KM.

Answer: In the revised manuscript, we have added a citation to this article in the introduction section.

3. The authors state: “For enzymes without obtainable UniProt IDs, an attempt was made to search for unique amino acid sequences using the EC number and organism name.” How was this done? Which data or database was used to achieve this? And if multiple sequences were found for the same organism-EC pair, which sequence was chosen?

Answer: Thank you for your comments. This approach actually originates from the DLKcat paper. The authors of DLKcat used the EC numbers and the organism names to retrieve amino acid sequences for cases where UniProt IDs were unknown when collecting kcat data. In our data collection, we also attempted this method to obtain enzyme sequences for unknown UniProt IDs. However, we ultimately did not include these sequences in the final dataset, as EC numbers and organism names alone were insufficient to identify a unique enzyme sequence. Therefore, all amino acid sequences included in our database were retrieved exclusively based

on UniProt IDs. We apologize for the omission of this critical information in the original manuscript. In the revised version, we have removed this sentence to prevent any potential misunderstanding for readers.

- 4. Supplementary Figure 1: The figure says “remove duplicates by SMILES”. What does this mean? I assume that the same SMILES occur many different data points. This should be made more precise.**

Answer: When collecting kcat and Km entries, we first compile details such as the enzyme EC number, enzyme type (wild-type or mutant), organism, UniProt ID, substrate name, and kcat or Km values. However, when SMILES are downloaded based on substrate names, multiple substrate names are sometimes found to correspond to the same SMILES. To ensure the uniqueness of each data point, this redundancy is removed.

- 5. Supplementary Figure 2: Annotation says “Keep maximum among among multiple measurements”, but for KM, I believe, it should be the minimum.**

Answer: We sincerely apologize for the oversight. As you noted, we did utilize the minimum value from multiple measurements when collecting the Km data. This error has been rectified in the revised manuscript.

- 6. “The SMILES string of the substrate was downloaded from PubChem[35] on the substrate name”. How many substrates could not be mapped to SMILES?**

Answer: Among all the kcat and Km entries collected from the BRENDA and SABIO-RK databases, a total of 30,691 unique substrate names were included, but 17,306 of these could not be assigned SMILES.

- 7. SMILES strings were used to represent small molecules? Were SMILES strings standardized, for example to canonical SMILES?**

Answer: Yes, small molecules are represented by canonical SMILES.

- 8. Typing errors: a. Line 287: “performs” instead of “preforms” b. It should be “MACCS keys fingerprint” instead of “MACC fingerprint” (l. 125). c. It should be “correction term” instead of “correlation term” (l. 459) d. Inconsistent writing: “machine learning” is written in lower case letters, whereas “Deep Learning” is written in upper case letters (line 61). e. Escherichia coli should not be written as one word (line 63)**

Answer: Thank you for your reminder. We have corrected these typing errors in the revised manuscript.

- 9. Ambiguous choice of words: In Figure 1, the authors refer to the final prediction modules based on the numerical representations as “decoder”. Since the pipeline involves a transformer network, I would suggest using a different word than “decoder” to make a clear distinction to transformer decoders.**

Answer: In Fig. 1, we replaced “decoder” with “predictor”.

- 10. Missing references: Partly, I felt that relevant references are missing. For example, in the**

paragraph from line 45 - 53, the authors described various methods or procedures without explicitly naming or referencing any.

Answer: Thank you for your reminder. We have added the corresponding references in the revised manuscript.

11. Some authors are referred to by their first names and not by their last names. For example, “Alexander et al.” Should be “Kroll et al.” and “David et al.” should be “Heckmann et al.”.

Answer: Thank you for your reminder. We have made the corrections in the revised manuscript.

12. The authors used an ESM-2 model, but do not state which of the many ESM-2 models was used.

Answer: The ESM2 version we used is esm2_t33_650M_UR50D, and we have clarified this point in the revised manuscript.

13. Extra tree was used as a prediction, but it is not specified which algorithm or package was used?

Answer: The extra tree method is imported from the scikit-learn package, and we have annotated this in the revised manuscript.

This article presents a novel approach by integrating enzyme kinetics and transfer learning, offering a new tool for enzyme discovery and directed evolution. Instead of using the traditional ratio of K_{cat} to K_m to predict catalytic efficiency (K_{cat}/K_m), the authors directly predict K_{cat}/K_m through a transfer learning model. This innovative method not only improves the accuracy of predictions but also simplifies the process, demonstrating significant potential for practical applications. The study validates the model's accuracy and utility through the discovery and modification of CSO enzymes, providing a fresh perspective and methodology for the field of enzyme engineering. Compared to existing kinetic models, the proposed transfer learning approach opens new avenues for handling large enzyme databases, with the potential to significantly impact the efficient discovery of novel enzymes and optimization of enzyme activity. Despite the many innovations in this article, there are some minor questions.

- 1. The first paragraph of the introduction only discusses the time and cost associated with obtaining high-activity enzyme variants, without adequately addressing the challenges in enzyme discovery. However, considering the discussion in section 2.7 of the results, it is recommended to include a discussion of the challenges commonly encountered in enzyme discovery within the introduction, as well as how artificial intelligence technology can significantly advance enzyme mining efforts. This would better align with the paper's focus on "enzyme discovery" as reflected in the title and provide a more comprehensive background for the readers.**

Answer: Thank you for your valuable suggestion. A major challenge in enzyme discovery is the limited availability of experimental data, which provides minimal reference for new enzyme discovery and engineering efforts. Huang et al. employed a strategy based on AF2 structure prediction and clustering protocols for the discovery of deaminases and subsequently designed the smallest single-strand-specific cytidine deaminase ([https://www.cell.com/cell/fulltext/S0092-8674\(24\)00718-9](https://www.cell.com/cell/fulltext/S0092-8674(24)00718-9)). Although these structure-based DL methods have proven effective for novel enzyme discovery, they may exhibit some randomness and chance when applied to certain specific substrates, particularly non-natural substrates. Therefore, we hypothesize that combining substrate-specific enzyme activity prediction models with protein structure clustering-based enzyme discovery strategies could enhance both efficiency and specificity. Based on your suggestion, we have added this to the introduction section of the revised manuscript.

- 2. The transition between the second and third paragraphs of the introduction is weak, making the structure feel disjointed. It is suggested to introduce transition sentences between these paragraphs to enhance the logical connection and create a smoother flow, allowing readers to more easily follow the background and significance of the research.**

Answer: Following your suggestions, we have revised the first few paragraphs of the introduction.

- 3. Please ensure consistent decimal places in the figures. For example, in Figures 2 and 3, the number of decimal places varies.**

Answer: Thank you for your reminder. We have modified the figure to ensure consistency in the number of significant decimal places.

4. **The data set in Fig. 3a shows that k_{cat}/K_m values exhibit a normal distribution. This raises two concerns: first, if a given enzyme's k_{cat}/K_m falls outside this distribution, can the model's predictions still maintain accuracy? Second, based on this distribution, does this imply that the prediction is only accurate when the results fall within this range?**

Answer: Predicting enzymatic reactions with experimental k_{cat}/K_m values outside this distribution range is indeed challenging. One of the main challenges AI faces in biological applications is the lack of rich data. For example, AlphaFold2's prediction accuracy significantly drops when high-quality templates are unavailable. This indicates that such issues are widespread across various tasks. Alternatively, the current experimental k_{cat}/K_m data are mostly concentrated between -3 and 3 ($\log_{10}[s^{-1} \cdot mM^{-1}]$), indicating that the k_{cat}/K_m values of the majority of reactions of interest are within this range. Hence, the enzymatic reactions encountered in new projects are likely to fall within this range as well. Based on the above analysis, for reactions with predicted k_{cat}/K_m values within the range of -3 to 3 ($\log_{10}[s^{-1} \cdot mM^{-1}]$), the predictions are generally more accurate. However, it cannot be ruled out that for reactions with true k_{cat}/K_m values outside this range, the predicted values might still fall within the -3 to 3 range, potentially leading to significant errors, although this is likely to be rare.

5. **Result 2.7 Apart from CSO enzymes, could the mining and modification of other more common enzymes be discussed to demonstrate the accuracy of the model? Additionally, are there sequences in the training set that are similar to the CSO enzymes?**

Answer: Thank you for your suggestion. Selecting additional enzymes for wet-lab experiments is indeed challenging for us, as it would require more time and funding. We apologize for this limitation. To demonstrate the potential applicability of our model, we have used experimental datasets from other sources as additional test sets in **Sections 2.5 and 2.6**. We appreciate your understanding of the challenges we are encountering.

The five selected CSO alternative enzymes, PpCSO, MgpCSO, PgCSO, SsCSO, and TkCSO, exhibit maximum sequence identities of 100.0%, 35.1%, 41.6%, 39.0%, and 43.1%, respectively, with our dataset. In the revised manuscript, we added the description "Only PpCSO is included in the CataPro dataset, while the maximum sequence identity between the other four enzymes and those in our dataset is below 44%" in Section 2.7 to clarify the similarity relationship between these enzymes and those in the training set.

6. **Line 304. In the CSO mining process, sequences with a length difference higher than 20 amino acids from CSO2 were excluded. What is the basis for this deletion? This approach may have led to the loss of potentially higher-activity sequences.**

Answer: Although excluding proteins with a length difference of more than 20 amino acids from CSO2 may lead to the omission of some highly active alternative enzymes, our goal is to ensure that the candidate enzymes exhibit greater overall similarity to CSO2, rather than just partial structural similarity, thereby reducing the likelihood of false positives.

7. **Fig. 5. The authors compare the predicted values of the mutant enzymes with the experimental values of the wild-type enzyme to demonstrate the model's accuracy.**

However, I believe this approach may introduce bias. The predicted activity of the wild-type enzyme should be used as the reference instead of the experimental values. If only predicted values are used for comparison, will the results still adhere to expected trends?

Answer: In **Fig. 5**, the values on the y-axis represent the predicted k_{cat}/K_m values from CataPro, as you noted.

The manuscript "Robust Enzyme Discovery and Engineering with Deep Learning using CataPro" by Wang et al. presents a fair and unbiased dataset to evaluate models predicting enzymatic kinetic parameters, as well as a framework for predicting these parameters. The question addressed is important and likely of interest to a broad audience. The authors conducted many experiments (comparison of different methods, architecture choices...) with several datasets (k_cat, Km, kcat/Km, DMS...) in different conditions (biased, unbiased...). The code is available for all in open access. However, I do have major concerns regarding the original contribution with respect to previous work and the comparison with SOTA.

- The position with respect to UniKP (ref 17, at least one author in common with the present manuscript) is not clearly stated. Specifically the authors claim that "Unlike previous studies, we also conducted the first evaluation of the capability of the enzyme kinetic parameter prediction model to identify mutation effects." but this does not seem true since mutation effects were already predicted in Ref 17. The paragraph and results about "extra trees" are not clear. The authors show that performance of CataPro improves with extra trees. I guess they replaced the linear layer with extra trees, is it the case? The authors' claim that they are using extra trees "instead of neural networks" is very confusing. Then, the authors show that CataPro outperforms UniKP(*) on Fig. 2 but that CataPro (extra trees) has performance similar to UniKP on Supp Fig. 8 (for instance). What has changed between these two experiments? Does the change come from the dataset? Or did the authors initially replace the extra tree in UniKP with a linear? And if the performance are similar, the contribution of CataPro to the state of the art for predicting kcat and Km is questionable.

Answer: Thank you for your comments. In the revised manuscript, we changed the sentence "Unlike previous studies, we also conducted the first evaluation of the capability of the enzyme kinetic parameter prediction model to identify mutation effects." to "Additionally, we also evaluated the ability of CataPro and other baseline kinetic parameter models to identify enzyme mutation effects using the ten-fold unbiased cross-validation dataset" to more clearly highlight our contribution.

Previous enzyme kinetic prediction models typically employed random splitting of training and test sets, which can lead to overly optimistic performance estimates that may not reflect true predictive capabilities, as enzymes encountered in real-world applications are generally unfamiliar to the model. Therefore, our work highlights the importance of training and evaluating enzyme kinetic parameter prediction models on the unbiased datasets. According to **Fig. 2a-f**, CataPro shows clear advantages over UniKP on the unbiased ten-fold cross-validation datasets. In contrast, on the randomly split, biased ten-fold cross-validation dataset for kcat, UniKP achieves a PCC of 0.820, exceeding CataPro's PCC of 0.798, indicating that UniKP is more prone to overfitting on biased datasets (**Supplementary Figure 8**). When the linear layer in CataPro was replaced with extra trees, the PCC increased from 0.798 to 0.823, further indicating that extra trees exhibit a stronger fitting capability on biased datasets, thereby intensifying overfitting. **Fig. 2h** clearly shows the substantial difference in model evaluation results between the biased (blue bar) and unbiased (red bar) datasets. Therefore, the series of tests conducted in this section aim to emphasize the importance of training and evaluating the model on unbiased test datasets. Since the enzymes and reactions involved in the currently available enzyme kinetic parameter assays represent only a trivial portion of those in

nature, overfitting to the existing data is meaningless. According to **Fig. 2a-f** and **Fig. 3d-f**, CataPro clearly outperforms UniKP on the unbiased datasets, achieving state-of-the-art performance.

- More generally, the manuscript lacks specific references to support sentences starting with "Researchers rely on co-evolution...", "some researchers employed..."

Answer: Thank you for your reminder. We have added the relevant references in the revised manuscript.

- The methods lacks information. How many layers? trainable parameters? What is precisely the architecture of the "decoding module based on attention mechanism"? Is it transformer-based attention?

Answer: The layers used in our kcat and Km models have been provided in the original manuscript, consisting of one hidden layer and one output layer. The prediction process for the kcat/Km model is described in greater detail in **Sections 2.1 and 4.3** of the revised manuscript to enhance readability. The hyperparameter search space and the final parameters used for model training are presented in **Supplementary Table 6**. In **Fig. 1c**, we employed feature-wise attention instead of the multi-head attention from transformer architectures, as detailed in **Section 4.3** of the revised manuscript.

- What is the level of noise in the experimental data? The authors takes the max value in case there are multiple, but they do not give indication about the extent to which these multiple values are contradictory or consistent. See for instance these study: <https://doi.org/10.1021/acs.jcim.4c00049> on thermodynamic parameters noise across datasets.

Answer: Thank you for your suggestion. There are 3,490 kcat and 6,468 Km data points in our final datasets exhibiting multiple labels from the original BRENDA and SABIO-RK databases, accounting for 12.6% and 15.4% of the total kcat and Km samples, respectively. Although the paper you provided introduced several evaluation metrics, such as R^2 and Kendall τ , to assess the noise level, these are typically used to evaluate the agreement between two assays. They may not be applicable to our case, as some enzymatic reactions may have more than two assays in the original databases. Therefore, to more intuitively illustrate the label noise in enzymatic reactions with multiple labels, we computed the standard deviation of the multiple labels for each reaction as a measure of the noise level. The noise distribution of multi-label samples in the kcat and Km datasets is presented in **Supplementary Fig. 3g** and **Supplementary Fig. 3h**, respectively.

- The results on the DMS seem very poor. The SCC values are low compared to what can be seen on ProteinGym leaderboard: <https://proteingym.org/benchmarks>. Particularly, the beta-lactamase is the classical example where variant effect predictors reach very high Spearman Correlation coefficient.

Answer: Thank you for your comments. This is indeed an interesting question. The fitness data of enzymes obtained through deep mutational scanning reflects their evolutionary characteristics, including the functions and properties that have developed throughout evolution. Factors that influence enzyme fitness are diverse and include stability, folding efficiency, catalytic activity, binding specificity and affinity (<https://www.biorxiv.org/content/10.1101/2023.12.07.570727v1>). Protein language models combined with self-supervised learning have been shown to effectively learn co-evolutionary information from large protein datasets

<https://www.science.org/doi/10.1126/science.ade2574>,

<https://www.biorxiv.org/content/10.1101/2023.10.01.560349v1>). As you mentioned, the language models on the ProteinGym leaderboard perform exceptionally well in predicting the fitness of beta-lactamase, significantly surpassing CataPro. This suggests that the protein language model has effectively learned the evolutionary information of proteins. However, the enzyme kinetic parameters, including k_{cat} , K_m , or k_{cat}/K_m , are only a part of the factors influencing fitness, thus they are not completely correlated with fitness. CataPro, as an enzyme kinetic parameter prediction model, is trained by fitting k_{cat} , K_m , or k_{cat}/K_m data, and its predictions are actually deviated from the fitness. Furthermore, our dataset also includes enzymatic kinetic data for enzymes catalyzing non-natural substrates, which is typically different from the evolutionary direction of the enzymes. Therefore, comparing enzyme kinetic parameter prediction models with the current SOTA protein language models using fitness data may not be entirely fair. However, although enzyme kinetic parameters are not strictly correlated with fitness, exploring their potential in fitness prediction remains meaningful, with previous studies on enzyme kinetic parameter prediction having not addressed this issue. As shown in Fig. 6, CataPro outperforms UniKP and DLKcat across the three DMS datasets. We have added a description in the first paragraph of **Section 2.6** in the revised manuscript to clarify the difference between fitness and enzyme kinetic parameters.

We sincerely appreciate the constructive suggestions from Reviewer 1. To enhance readability, our latest responses are highlighted in blue font.

Major concerns

1. Omission of a test set: Using only 10-fold cross-validation without a separate test set can lead to slightly optimistic performance estimates. (i) If you tune your model based on the average cross-validation results, you may inadvertently optimize for those specific folds, leading to overfitting. (ii) Without a separate test set, you never evaluate your model on completely new, unseen data. This doesn't fully mimic real-world scenarios. Using a separate final test set helps address these issues by providing a truly independent evaluation of your model's performance on data it has never seen before. Since generating a separate and independent test set would require redoing the entire analyses, this is obviously not practical or necessary. Moreover, the authors later evaluate the model on additional experimental data, which partially fills the role of independent test data. However, the authors could at least state that the results from the mean cross-validation metrics may be slightly overoptimistic compared to what one would see for truly independent data.

Answer in the first round of review: Thank you for your reminder and suggestions. We have added the following sentence to the discussion section: “Additionally, although CataPro performs well on the unbiased ten-fold cross-validation dataset, it may still encounter challenges when applied to highly unfamiliar proteins, reactions, or other independent test sets.”, and highlighted it in blue.

Reviewer response: The authors' response does not fully address my concern. While what they have added to the manuscript is correct, it does not address the challenges of using a 10-fold CV as the sole evaluation procedure. The challenges encountered when applied to highly unfamiliar proteins, reactions, or other independent test sets is a problem that models typically independently of the evaluation procedure. I suggest that the authors explicitly state why a 10-fold CV evaluation is likely to lead to slight overfitting when the same 10 folds are also used for hyperparameter optimization (see my original concern for more information).

Answer: Thank you for your valuable suggestions. According to your suggestion, we have clarified in the revised manuscript that the 10-fold cross-validation evaluation may lead to overfitting when the same 10 folds are used for hyperparameter optimization. We have added the following sentence to the Discussion section: “Additionally, it is important to note that the results in Fig. 2 represent the models' performance on all validation sets during training on the ten-fold cross-validation dataset. Since the validation sets were used in hyperparameter optimization during training, the evaluation of CataPro may be slightly overoptimistic, and it may still encounter challenges when applied to highly unfamiliar proteins, reactions, or other independent test sets.”, and highlighted it in blue.

2. When CataPro and UniKP are trained on the same data for kcat and KM prediction, the differences in model performance are small (Figure 2h). This suggests that the main advancement of CataPro for kcat and KM prediction lies in the design of the datasets rather than in a novel model architecture. This should be emphasized in the manuscript.

Answer in the first round of review: Fig. 2h shows the performance of CataPro (extra tree) and UniKP on both unbiased and biased datasets, where CataPro (extra tree) means replacing linear layer with extra tree in CataPro. Since the only difference between UniKP and CataPro (extra tree) lies in the representation of the substrate, the superiority of CataPro's feature

combination can be seen in Fig. 2h, although the advantage is limited, as you mentioned. Furthermore, Fig. 2h provides a more intuitive illustration of the performance differences of the models on biased and unbiased datasets. Here, we urge researchers to focus on the model's performance on unbiased datasets to enable a more objective evaluation. In accordance with your suggestion, we have enhanced the discussion in Section 2.2 in the revised manuscript and highlighted it in blue.

Reviewer response: What the authors have added to the manuscript is the opposite of what I aimed at with my concern. They now added that the “difference between CataPro (extra tree) and UniKP demonstrates that the features used by CataPro are more effective than those used by UniKP”. However, when we look at Figure 2h (upper panel for kcat), the difference between CataPro and UniKP on the biased dataset for PCC and SCC is within a range of 0.005 and for the unbiased dataset it looks like the differences are within a range of 0.02. These differences might not even be statistically significant. I repeat my prior statement: “This suggests that the main advancement of CataPro for kcat and KM prediction lies in the design of the datasets rather than in a novel model architecture” This is not a bad finding since a clean dataset might be even more important than some tweaking to the architecture and representations, but I still argue this should be clearly stated in the manuscript.

Answer: Thank you for your valuable suggestions. In the revised manuscript, we removed the sentence “the difference between CataPro (extra tree) and UniKP demonstrates that the features used by CataPro are more effective than those used by UniKP”, and add the sentence “Therefore, the primary advantage of our proposed CataPro lies in utilizing unbiased datasets for training and evaluation, rather than relying on a more novel model architecture.” in Section 2.2 to emphasize the importance of using the unbiased datasets to enhance CataPro's performance.

3. CataPro does not use any reaction information. The turnover number kcat depends on the whole chemical reaction catalyzed by an enzyme and not only on one substrate. However, CataPro only uses information about one of the substrates that is part of the reaction. This limitation is not mentioned or discussed anywhere in the manuscript. Why is the entire reaction not encoded? Is this limitation important for the performance of the model? How does one choose a single substrate in the case of reactions with multiple reactions?

Answer in the first round of review: Thank you for your comments. We agree that reaction information is an important factor. We ultimately do not use reaction information, mainly due to the following concerns: (1) Collecting reaction information introduces additional conditions and constraints, which will reduce the sample size of the final dataset. (2) With limited sample size, it will be difficult to support the model in achieving additional functionalities, such as the ability to rank mutants. When predicting enzyme kinetic parameters for reactions involving multiple substrates, we can select a key substrate as input. In fact, in the BRENDA database, some entries may omit one of the two substrates, and the omitted substrate is typically a common, low-molecular-weight molecule, such as water, oxygen, or metal ions. As discussed in Section 2.7, 4-vinylguaiacol is evidently a key substrate in the reaction.

Reviewer response: I disagree: The authors write “some [BRENDA] entries may omit one of the two substrates, and the omitted substrate is typically a common, low-molecular weight molecule, such as water, oxygen, or metal ions.”, but this is not the case. In multi-substrate reactions, to measure kcat, the concentration of all but one substrate is kept fixed (and high)

and the concentration of the remaining substrate is increased. BRENDA lists this substrate for which the concentration is increased. This may or may not be the key substrate. For example, if we look at the following BRENDA entry ([https://www.brenda.enzymes.org/enzyme.php?ecno=1.1.1.10#TURNOVER%20NUMBER%20\[1/s\]](https://www.brenda.enzymes.org/enzyme.php?ecno=1.1.1.10#TURNOVER%20NUMBER%20[1/s])), we see kcat values for the main substrate and for the co-factors. Thus, the CataPro dataset has many substrate entries where the substrate is not the main/key substrate. This is a major limitation. In addition, it would be possible to obtain the reaction equations for the majority of the data points with substantial additional work. I don't expect the authors to do this work, but to say that it is not possible is not accurate. The authors should clearly state this limitation and the potential problems it causes. It is also unclear to me how to obtain kcat predictions for new enzymatic reactions. Often it is not possible to define a key substrate, or when predictions are required for genome-scale metabolic models, it is impractical to hand-label all key substrates. It should be discussed how the model should be applied to obtain appropriate predictions.

Answer: Thank you for your valuable suggestion. In the revised manuscript, we have added “Another limitation is that CataPro predicts enzyme kinetic parameters for an enzyme and a single substrate, whereas many reactions involve multiple substrates. This means that CataPro relies on partial reaction information rather than complete reaction information, which may introduce the risk of systematic errors due to missing data. Therefore, collecting more data points that include comprehensive reaction information to train a model based on full reaction details remains a very promising direction.” to the Discussion section. This clarifies the limitation that CataPro does not utilize complete reaction information and the potential issues it may cause. As you mentioned, "it is impractical to hand-label all key substrates," it is indeed challenging to manually label key substrates for all reactions in genome-scale enzymatic reaction predictions. In the revised manuscript, we tested an alternative approach using the kcat dataset containing complete reaction information, developed by Kroll et al. (TurNup paper). Specifically, for reactions involving multiple substrates, we first used CataPro to predict the kcat for each enzyme-substrate pair, and then took the average of all enzyme-substrate kcat values as the final kcat for the reaction. On the test set (850 data points), CataPro achieved an R of 0.661 and an R² of 0.415, which are very close to the R of 0.67 and R² of 0.44 obtained by TurNup, while there was a large gap between CataPro and TurNup in the 0–40% subset, where the sequence identity between enzymes in the TurNup test set and CataPro kcat dataset was below 40%. We have discussed these results in detail in Concern-4.1 you mentioned and Section 2.2 of the revised manuscript.

- 4.1 Comparison with TurNuP and KM prediction model: The authors did not perform a comparison with the TurNuP model, which showed much better performance than DLKcat. While the TurNuP model architecture is not special, the authors put a lot of effort into data cleaning and preprocessing, such as determining the complete chemical reaction equations for all data points. Since TurNuP and CataPro were evaluated on different datasets, a direct fair comparison is not possible. However, if we consider (i) in both cases only test enzymes with a sequence identity of less than 40% compared to the training enzymes and (ii) that the PCC squared is approximately equal to the coefficient of determination R², we obtain the following comparison: CataPro PCC²=(0.497)²=0.247 vs. TurNuP R² = 0.328 Performing the same comparison for CataPro and the ProSmith Km (<https://doi.org/10.1371/journal.pcbi.1012100>, see Minor Concern 2.) prediction model for test sequences with less than 40% sequence identity, we obtain the following comparison: CataPro PCC²=(0.633)²=0.401 vs. ProSmith R² = 0.41 Can the

authors comment on why the results (at least in this simplified comparison) do not show significant differences between the two prediction models? TurNuP and ProSmith_KM only provide predictions for wild-type enzymes? Could it be that both models perform similarly for wild-type enzymes?

Answer in the first round of review: Thank you for your suggestion. To ensure a fair comparison with TurNuP, we trained and tested our model on the dataset used by TurNuP. Since the TurNuP dataset includes all substrates and products for each reaction, we extracted feature vectors for each substrate and product individually. For reactions with multiple substrates or products, we calculated the average of the corresponding feature vectors to serve as the final representation of the substrates or products. Then, the feature vectors of the enzyme, substrate, and product are concatenated to obtain the representation of the entire reaction (Supplementary Fig. 9a). Finally, a linear layer, as used in Fig. 1b, is applied to predict kcat. We trained CataPro on the five-fold cross-validation dataset divided by Kroll et al., and then evaluated the resulting five models on the four subsets of the test set, which were divided based on the enzyme's maximum sequence identity to those in the training set. The results are shown in Supplementary Fig. 9b. An interesting observation is that one of the models (Fold-5) achieved an R^2 of 0.374 in the subset with the maximum sequence identity below 40%, slightly outperforming 0.33 of TurNuP. It is important to note that the number of training samples for each sub-model of CataPro is actually four-fifths of the TurNuP training samples, as one-fifth of the samples are used for validation. Another notable observation is that, within the subset exhibiting a maximum sequence identity of 99%-100%, the performance of the five CataPro models varies considerably. This discrepancy could be attributed to the fact that each training set is missing one-fifth of the samples from the original dataset. We discussed these results in Section 2.3 of the revised manuscript.

Reviewer response: Concerns regarding the comparison: (i) The CataPro model does not utilize reaction information, because no training dataset with full reaction information was assembled. However, for the comparison with TurNuP, the authors included reaction information from the TurNuP dataset. Therefore, this version of CataPro uses more information than the original CataPro version and therefore does not show the differences between TurNuP and the presented CataPro model, which does not use full reaction information. (ii) If we average the results from the 5-fold CV, CataPro is no better than TurNuP (maybe a little worse) across the different sequence similarities. However, the authors have simply picked the only one of the 5 folds where the model performs better (on the 0-40% similarity subset) on the test set, and they only report the result of that specific fold. This is not a fair comparison at all. If the authors are concerned about limited training set size, they can do the same as TurNuP and do hyperparameter optimisation on the 5-fold CV, pick a set of hyperparameters, and then train the model on the entire training set with the pre-selected hyperparameters. (iii) A more general comment about all comparisons: A prediction model is defined not only by its model architecture, but also strongly by its training data and training process. However, the authors do not make these distinctions. For example, here the authors make a comparison between the TurNuP and CataPro architectures on the TurNuP dataset, which means that this comparison does not show the performance of the CataPro model presented in this study, which was trained on different data. I think it is important to make this distinction clear in the manuscript.

Answer: Thank you for your valuable suggestion. According to your comment that “A prediction model is defined not only by its model architecture, but also strongly by its training

data and training process,” we directly tested CataPro, which was trained on our kcat ten-fold cross-validation dataset, on the TurNuP test set curated by Kroll et al. For multi-substrate reactions, we first predicted the kcat values for the enzyme in combination with each substrate and then took the average of these kcat values as the final prediction for the reaction. Ultimately, CataPro achieved an R value of 0.661 and an R^2 of 0.415, which are very close to TurNuP’s R of 0.67 and R^2 of 0.44. This suggests that, for multi-substrate reactions or cases where identifying the key substrate is challenging, this approach could potentially serve as an alternative. Following the method of Kroll et al., we partitioned the test set into four subsets based on the maximum sequence identity to enzyme sequences in the CataPro kcat dataset: 0–40%, 40–80%, 80–99%, and 99–100%. The results show that CataPro’s performance in the 0–40% subset exhibits a significant gap compared to TurNuP (Supplementary Fig. 9a).

Supplementary Figure 9. a The performance of CataPro on the TurNuP test set across four subsets, which were divided based on the maximum sequence identity to enzyme sequences in the CataPro k_{cat} dataset. **b** The performance of the retrained CataPro on the four subsets of the TurNuP test set, which are consistent with those in the original TurNuP paper. The number of samples in each subset is shown above each point.

This finding may also align with the concern you raised in “comment 1,” suggesting that evaluating the model solely on the 10-fold cross-validation dataset could lead to overly optimistic results. We have reported this limitation in the revised manuscript to provide a more comprehensive assessment of CataPro. Moreover, based on your comments in (i) and (ii), we retrained CataPro using the TurNuP training set compiled by Kroll et al. For each multi-substrate reaction, the final substrate feature was represented as the average of all substrate features. The training framework was consistent with that shown in Fig. 1b, which included only enzyme and substrate information. The retrained CataPro achieved an R value of 0.672 and an R^2 of 0.451, comparable to TurNuP’s R of 0.67 and R^2 of 0.44. In the four subsets divided based on the maximum sequence identity to enzyme sequences in the training set, the retrained CataPro exhibited performance similar to TurNuP (Supplementary Fig. 9b). This may suggest that CataPro’s architecture is comparable to TurNuP’s architecture in handling complex enzymatic reactions when trained on datasets containing full reaction information. This also provides a new perspective for further developing kcat models based on datasets that incorporate full reaction information. We hope this revision adequately addresses your concerns.

4.2 Answer in the first round of review: ProSmith is also an interesting work. The similar performance of ProSmith_KM and CataPro_KM based on your rough comparison might also suggest that the differences among current models for Km prediction are perhaps not significant, as shown in Fig. 2, at least smaller than the differences in kcat prediction.

Reviewer response: The authors do not discuss this equal performance between ProSmith_Km and CataPro at all in the manuscript. As there is already a KM prediction model with similar prediction capabilities, this should be discussed.

Answer: Thank you for your valuable suggestion. ProSmith utilized ESM-1b and ChemBERTa2 to tokenize protein and small molecule sequences, respectively. It then employed a transformer network for pretraining, enabling the custom <cls> token to encode protein-ligand interaction information. In addition, ProSmith also incorporated embeddings from ESM-1b and ChemBERTa2 as supplementary features. Therefore, we speculate that the similar performance of ProSmith and CataPro in the Km prediction task may be due to their use of transformer-based pretrained models to generate enzyme-substrate representations. Moreover, the Km datasets for both CataPro and ProSmith are derived from the BRENDA and SABIO-RK databases, which might be one of the reasons for the comparable performance of these two models. We have added a discussion in Section 2.2.

5. Analysis in section 2.4: If I understand correctly, the authors evaluated the performance for the mutant kcat prediction for different enzyme families simultaneously. However, this can potentially lead to an overly optimistic PCC as the following simple example demonstrates: Suppose that we have three wildtype enzymes, W1, W2 & W3, with kcat values of 1/s, 100/s, and 1000/s, respectively. We now want to predict the kcat value for three mutants of these wildtypes, M1, M2 & M3, with true kcat values that are significantly different from the corresponding wildtype kcat values: 0.1/s, 130/s and 1300/s, respectively. If we have a model that simply predicts the kcat values of the wild-type enzymes and thus predicts values close to 1/s, 100/s and 1000/s for the mutants, the model clearly cannot predict the effect of the mutations on the kcat value. However, if we compare the raw predicted values with the raw true values for the mutants, we still obtain a very high PCC value due to the large scale of the kcat values, and it appears as though we have an almost perfect mutant prediction model (see figure below). The accuracy of identifying advantageous mutations could also be slightly biased if we assume that most mutants lower the kcat value. A more intuitive and unbiased measure could be to compute for each mutant the difference compared to the corresponding wildtype kcat value and then comparing predicted differences from true differences. Such delta values as computed in <https://doi.org/10.1093/biomethods/bpae061> can then be compared across different enzymes.

Answer in the first round of review: In fact, what we are evaluating here is the model's ability to rank mutants for each reaction (with the same UniProt ID and SMILES). For example, in Fig. 4a-c, we first calculated the SCC achieved by the model for each reaction, and then presented the SCC results across all reactions in the form of a box plot. Therefore, the method you mentioned is actually consistent with the approach used in Fig. 4a-c of our manuscript. Fig. 4d-f demonstrates the model's ability to predict advantageous and disadvantageous mutants. Taking Fig. 4a as an example, advantageous mutants are defined as those where the real kcat value is higher than that of the wild-type. If the model's predicted kcat for this mutant is greater than that for the wild-type, it indicates a successful prediction. We calculated the accuracy of the model for each reaction (with the same UniProt ID-Smiles) and presented the model's performance across all reactions in the form of a box plot. Therefore, these two metrics (SCC and Accuracy) can provide a more intuitive assessment of the model's ability to rank mutants for the specific reaction.

Reviewer response: The ranking capabilities of the resulting models for this difficult

prediction task do not appear to be very high. In Figure 4a-c, the average SCC for $N \geq 30$ seems to be ≤ 0.2 , i.e. even if more than 30 mutants for the same enzyme were in the training set, the model cannot accurately rank unseen mutants. I was therefore surprised to see an accuracy of 70% in Figure 4d-f. I think it is likely that there is a bias in the dataset where, for example, for one enzyme, many of the reported mutant values are all higher or lower than the wildtype *k_{cat}* values. To test if such a bias exists and if the model can actually predict something other than a bias, the authors could normalise the predictions and the true values by subtracting the mean of all values of the same enzyme-SMILES pair in the training set. If the model only learns to predict an average bias for all mutants of a particular enzyme-SMILES pair, the predictions will be close to that average.

Answer: We greatly appreciate your further suggestions and guidance. In Section 2.4, reactions (enzyme-SMILES pairs) are used for evaluation only when they are included in the validation set. Taking *k_{cat}* as an example, since the models were trained on the unbiased 10-fold cross-validation dataset, if a particular enzyme-substrate pair appears in the validation set, the training set will not contain any enzymes with a sequence similarity greater than 0.4 to that enzyme. Therefore, Section 2.4 essentially evaluated the models' ability to rank mutants of unseen enzymes. Based on your comment, "I think it is likely that there is a bias in the dataset where, for example, for one enzyme, many of the reported mutant values are all higher or lower than the wildtype *k_{cat}* values," we have conducted a statistical analysis of the effects of mutants relative to the wildtype for each reaction (same enzyme-substrate pair). As you mentioned, most mutants show a worse effect (Supplementary Figs. 11d-f). Therefore, the analysis distinguishing advantageous and disadvantageous mutations relative to the wildtype, which we previously presented in Figs. 4d-f and Figs. 4g-i, indeed contains a bias. In the revised manuscript, we have removed these two sets of figures. In addition, we have adopted a new method to unbiasedly analyze the models' ability to identify "better-performing" mutations: first, for a reaction (same enzyme-substrate) with $N \geq 20$ or $N \geq 30$, we paired all mutants in a pairwise combination, resulting in $N(N-1)/2$ pairs. Then, the models were used to rank each pair of mutants, and finally, we calculated the accuracy of the model in correctly ranking the pairs for that reaction. The results are presented in Figs. 4d-f of the revised manuscript.

Finally, based on your comment "To test if such a bias exists and if the model can actually predict something other than a bias, the authors could normalise the predictions and the true values by subtracting the mean of all values of the same enzyme-SMILES pair in the training set" as well as your suggestions in the first round (including the reference <https://doi.org/10.1093/biomethods/bpae061>), we conducted a global analysis of the models' ability to predict mutation effects. Taking *k_{cat}* as an example, for each enzyme-substrate pair, we first calculated the mean of the measured *k_{cat}* values for all mutants (including the wild type). Then, both the predicted and true values of all mutants subtracted this mean to obtain the predicted mutation effect and the true mutation effect. We found that CataPro exhibits excellent correlation between predicted and experimental mutation effects for certain reactions, despite the training set containing no enzymes with more than 40% similarity to the enzyme (Supplementary Fig. 12a-c). Conversely, for some reactions, CataPro demonstrates weaker correlations (Supplementary Fig. 12d-f). Notably, even for reactions with high correlation, CataPro demonstrates limitations in accurately predicting the absolute values of mutation effects.

Supplementary Figure 12. Predictions of CataPro in representative enzymatic reactions. **a**, **b**, and **c** show the enzymatic reactions with the highest correlation in predicting mutation effects by CataPro on the k_{cat} , K_m , and k_{cat}/K_m datasets, with the corresponding enzyme (UniProt ID)-substrate pairs being (Q9UKK9, ADP-D-ribose), (P50384, Anthranilate), and (Q9UKK9, ADP-D-ribose), respectively. **d**, **e**, and **f** show the enzymatic reactions with the lowest correlation in predicting mutation effects by CataPro on the k_{cat} , K_m , and k_{cat}/K_m datasets, with the corresponding enzyme (UniProt ID)-substrate pairs being (P13956, S-Adenosyl-L-methionine), (P26276, alpha-D-Glucose 1-phosphate), and (P51570, ATP), respectively.

We also conducted a global analysis by evaluating all reactions, but neither CataPro, UniKP, nor DLKcat successfully predicted the mutation effects (Supplementary Table 3). This may suggest that accurately predicting enzyme mutation effects remains a considerable challenge. We sincerely appreciate your valuable suggestions once again, which have been very helpful in improving our manuscript.

6. How did the authors handle multimer enzymes with multiple different UniProt IDs? Which of the UniProt IDs was chosen in those cases?

Answer in the first round of review: During data collection, we excluded samples with multiple UniProt IDs.

Reviewer response: If I am not mistaken, this is not mentioned in the manuscript, but this is important information that should be included.

Answer: Thank you for your reminder. We have added this information in Section 4.1 of the revised manuscript and highlighted it in blue.

Minor concerns

1. “The SMILES string of the substrate was downloaded from PubChem[35] on the substrate name”. How many substrates could not be mapped to SMILES?

Answer: Among all the kcat and Km entries collected from the BRENDA and SABIO-RK databases, a total of 30,691 unique substrate names were included, but 17,306 of these could not be assigned SMILES.

Reviewer response: This information should be included in the manuscript

Answer: We have added this information in Section 4.1 of the revised manuscript and highlighted it in blue.

2. SMILES strings were used to represent small molecules? Were SMILES strings standardized, for example to canonical SMILES?

Answer: Yes, small molecules are represented by canonical SMILES.

Reviewer response: This information should be included in the manuscript

Answer: We have added this information in Section 4.1 of the revised manuscript and highlighted it in blue.

GENERAL COMMENTS

The manuscript "Robust Enzyme Discovery and Engineering with Deep Learning using CataPro" presents a new method for predicting enzyme turnover numbers (k_{cat}), Michaelis constants (K_M), and catalytic efficiencies (k_{cat}/K_M). The novelty of the method is mainly based on a novel approach for predicting k_{cat}/K_M and an appropriate and careful partitioning of the data sets into training and validation sets. The authors claim that their method outperforms other state-of-the-art models for the same prediction tasks and is able to predict the effects of mutations. The results of the experiments performed to improve the catalytic activity of enzymes are indeed impressive.

As detailed in the Major Concerns below, my major concerns relate to missing information and comparisons with previous methods. Although I have many major concerns, I believe that the study makes an important contribution by showing (i) that careless data set splitting only leads to seemingly high model performance and (ii) that current models for predicting kinetic parameters can play an important role in real-world enzyme engineering tasks. I believe that my concerns can be addressed in a major revision by clarifying or adapting the methods and by providing much more required information within the manuscript.

MAJOR CONCERNS

1. In several places the authors discuss the limitations of DLKcat (e.g. ll. 66ff., 80ff., l.351). There is a recent publication that discusses the limitations of DLKcat: <https://doi.org/10.1093/biomethods/bpae061>. This publication should be cited in these places.
2. Omission of a test set: Using only 10-fold cross-validation without a separate test set can lead to slightly optimistic performance estimates. (i) If you tune your model based on the average cross-validation results, you may inadvertently optimize for those specific folds, leading to overfitting. (ii) Without a separate test set, you never evaluate your model on completely new, unseen data. This doesn't fully mimic real-world scenarios. Using a separate final test set helps address these issues by providing a truly independent evaluation of your model's performance on data it has never seen before. Since generating a separate and independent test set would require redoing the entire analyses, this is obviously not practical or necessary. Moreover, the authors later evaluate the model on additional experimental data, which partially fills the role of independent test data. However, the authors could at least state that the results from the mean cross-validation metrics may be slightly overoptimistic compared to what one would see for truly independent data.
3. When CataPro and UniKP are trained on the same data for k_{cat} and K_M prediction, the differences in model performance are small (Figure 2h). This suggests that the main advancement of CataPro for k_{cat} and K_M prediction lies in the design of the datasets rather than in a novel model architecture. This should be emphasized in the manuscript.
4. CataPro does not use any reaction information. The turnover number k_{cat} depends on the whole chemical reaction catalyzed by an enzyme and not only on one substrate. However, CataPro only uses information about one of the substrates that is part of the reaction. This limitation is not mentioned or discussed anywhere in the

manuscript. Why is the entire reaction not encoded? Is this limitation important for the performance of the model? How does one choose a single substrate in the case of reactions with multiple reactions?

5. kcat/KM prediction architecture:

- a. The authors write “this approach [directly predicting kcat/KM] may not achieve optimal performance, as it makes kcat/KM entirely independent of kcat and KM” (l. 130f.). Did the authors test whether their approach actually leads to superior performance compared to a simple model that directly predicts kcat/KM with a single model?
- b. The authors apply “an attention layer before the output layer [that] extracts crucial information” (l.138). The attention layer is applied to a single vector, right? How is the attention function applied to a single vector? Can the authors give a function?
- c. For the kcat/KM prediction model, models were pre-trained to predict only kcat and only KM. What data were these models trained on? The authors state “we pre-train the kcat and Km models on the tenfold cross-validation dataset for kcat/KM”, but the kcat/Km dataset does not contain kcat and KM values. Does this mean that the models are pre-trained on KM and kcat data for enzymes in the kcat/KM training folds for which you can find values in the kcat and KM datasets?
- d. The exact pipeline and architecture of the kcat/KM prediction model is unclear from the main text and methods. It seems that the KM and kcat models are first pre-trained and then fixed, right? Is a model then trained to predict the differences between the predicted kcat/KM (with the above models) and the actual kcat/KM? It should be described in detail when and how which model is trained and how the final prediction is achieved.

6. Comparison with TurNuP and KM prediction model: The authors did not perform a comparison with the TurNuP model, which showed much better performance than DLKcat. While the TurNuP model architecture is not special, the authors put a lot of effort into data cleaning and preprocessing, such as determining the complete chemical reaction equations for all data points. Since TurNuP and CataPro were evaluated on different datasets, a direct fair comparison is not possible. However, if we consider (i) in both cases only test enzymes with a sequence identity of less than 40% compared to the training enzymes and (ii) that the PCC squared is approximately equal to the coefficient of determination R², we obtain the following comparison:

CataPro $PCC^2 = (0.497)^2 = 0.247$ vs. TurNuP $R^2 = 0.328$

Performing the same comparison for CataPro and the ProSmith Km (<https://doi.org/10.1371/journal.pcbi.1012100>, see Minor Concern 2.) prediction model for test sequences with less than 40% sequence identity, we obtain the following comparison:

CataPro $PCC^2 = (0.633)^2 = 0.401$ vs. ProSmith $R^2 = 0.41$

Can the authors comment on why the results (at least in this simplified comparison) do not show significant differences between the two prediction models? TurNuP and ProSmith_KM only provide predictions for wild-type enzymes? Could it be that both models perform similarly for wild-type enzymes?

7. Differences in prediction performance between KM and k_{cat} . The authors give the following reasons for the differences in model performance between KM and k_{cat} prediction: “These results indicate that predicting k_{cat} is more challenging than predicting KM. This is because, in certain enzyme-catalyzed reactions, single-point mutations of enzymes may result in k_{cat} values varying by several orders of magnitude. However, mutated enzyme representations often exhibit limited feature differences, especially among similar mutants. Another factor is that the current amount of k_{cat} data is small compared to the vast mutation space of enzymes, making it difficult for models to learn the impact of mutations on k_{cat} [39].” I find the arguments not very convincing. Mutations can also cause large differences in KM (and differences can also be seen in the wildtype-only models), and I believe the difference in dataset sizes are not big enough to explain the large differences in model performance. My suspicion would be that (i) it is simply a more difficult task to predict k_{cat} than KM, and (ii) k_{cat} measurements can be more noisy than KM measurements, and hence training data for k_{cat} might be more noisy.
8. Analysis in section 2.4:
- Were all mutants and wildtypes analyzed in this section always part of the training set?
 - If I understand correctly, the authors evaluated the performance for the mutant k_{cat} prediction for different enzyme families simultaneously. However, this can potentially lead to an overly optimistic PCC as the following simple example demonstrates:
Suppose that we have three wildtype enzymes, W1, W2 & W3, with k_{cat} values of 1/s, 100/s, and 1000/s, respectively. We now want to predict the k_{cat} value for three mutants of these wildtypes, M1, M2 & M3, with true k_{cat} values that are significantly different from the corresponding wildtype k_{cat} values: 0.1/s, 130/s and 1300/s, respectively. If we have a model that simply predicts the k_{cat} values of the wild-type enzymes and thus predicts values close to 1/s, 100/s and 1000/s for the mutants, the model clearly cannot predict the effect of the mutations on the k_{cat} value. However, if we compare the raw predicted values with the raw true values for the mutants, we still obtain a very high PCC value due to the large scale of the k_{cat} values, and it appears as though we have an almost perfect mutant prediction model (see figure below):

The accuracy of identifying advantageous mutations could also be slightly biased if we assume that most mutants lower the k_{cat} value. A more intuitive and unbiased measure could be to compute for each mutant the difference

compared to the corresponding wildtype kcat value and then comparing predicted differences from true differences. Such delta values as computed in <https://doi.org/10.1093/biomethods/bpae061> can then be compared across different enzymes.

9. Analyses in Sections 2.5 and 2.6: Did the authors make sure that none of the sequences from the external databases were present in the CataPro dataset to avoid data leakage?
10. The description of how enzyme candidates were selected during the directed evolution described in Section 2.7 is unclear to me. The authors write “considering the PSSM scores, six representative mutants were selected for experimental validation”, which is vague. Can the authors give specific rules that they followed to select candidates to make the process understandable and reproducible?
11. An overall comment on the Methods section, which is related to some of my other concerns: The descriptions are very short and partly vague. I think that not enough information is given to successfully reproduce the study.
12. How did the authors handle multimer enzymes with multiple different UniProt IDs? Which of the UniProt IDs was chosen in those cases?
13. If I see it correctly, the training and validation datasets are not provided. Without these the study is not reproducible.
14. How were hyperparameters of the tested models optimized? Which hyperparameters were tested?

MINOR CONCERNS

1. Discussion of EITLEM paper: On 19.09.2024 the EITLEM paper was published, a new state-of-the art method for predicting kcat, Km and kcat/KM (full paper: <https://doi.org/10.1016/j.checat.2024.101094>, short preview article: [/www.cell.com/chem-catalysis/fulltext/S2667-1093\(24\)00308-7](https://www.cell.com/chem-catalysis/fulltext/S2667-1093(24)00308-7)). Since the paper was published after the CataPro authors have submitted their manuscript, I think the EITLEM does not need to be discussed or analyzed. However, the authors could add a short paragraph to the discussion, showing their awareness of recent advances and explaining that due to overlaps in the publication timeframes, no comparison was possible. But, as I said I believe this is optional and I just wanted to raise the author’s awareness to the similar study.
2. The authors refer to a KM prediction (DOI: [10.1371/journal.pbio.3001402](https://doi.org/10.1371/journal.pbio.3001402)), but this was recently improved in another publication (DOI: [10.1371/journal.pcbi.1012100](https://doi.org/10.1371/journal.pcbi.1012100)). I hence suggest, the authors cite the new, improved model for predicting KM.
3. The authors state: “For enzymes without obtainable UniProt IDs, an attempt was made to search for unique amino acid sequences using the EC number and organism name.” How was this done? Which data or database was used to achieve this? And if multiple sequences were found for the same organism-EC pair, which sequence was chosen?

4. Supplementary Figure 1: The figure says “remove duplicates by SMILES”. What does this mean? I assume that the same SMILES occur many different data points. This should be made more precise.
5. Supplementary Figure 2: Annotation says “Keep maximum among among multiple measurements”, but for KM, I believe, it should be the minimum.
6. “The SMILES string of the substrate was downloaded from PubChem[35] on the substrate name”. How many substrates could not be mapped to SMILES?
7. SMILES strings were used to represent small molecules? Were SMILES strings standardized, for example to canonical SMILES?
8. Typing errors:
 - a. Line 287: “performs” instead of “preforms”
 - b. It should be “MACCS keys fingerprint” instead of “MACC fingerprint” (l. 125).
 - c. It should be “correction term” instead of “correlation term” (l. 459)
 - d. Inconsistent writing: “machine learning” is written in lower case letters, whereas “Deep Learning” is written in upper case letters (line 61).
 - e. Escherichia coli should not be written as one word (line 63)
9. Ambiguous choice of words: In Figure 1, the authors refer to the final prediction modules based on the numerical representations as “decoder”. Since the pipeline involves a transformer network, I would suggest using a different word than “decoder” to make a clear distinction to transformer decoders.
10. Missing references: Partly, I felt that relevant references are missing. For example, in the paragraph from line 45 - 53, the authors described various methods or procedures without explicitly naming or referencing any.
11. Some authors are referred to by their first names and not by their last names. For example, “Alexander et al.” Should be “Kroll et al.” and “David et al.” should be “Heckmann et al.”.
12. The authors used an ESM-2 model, but do not state which of the many ESM-2 models was used.
13. Extra tree was used as a prediction, but it is not specified which algorithm or package was used?

General comments: Many of the major concerns were not fully or adequately addressed. In these cases, I have copied my original concern along with the authors' response, before responding with my new response, introduced by "Reviewer response:" and intended to make the document more readable.

Major concerns

- 1. Omission of a test set: Using only 10-fold cross-validation without a separate test set can lead to slightly optimistic performance estimates. (i) If you tune your model based on the average cross-validation results, you may inadvertently optimize for those specific folds, leading to overfitting. (ii) Without a separate test set, you never evaluate your model on completely new, unseen data. This doesn't fully mimic real-world scenarios. Using a separate final test set helps address these issues by providing a truly independent evaluation of your model's performance on data it has never seen before. Since generating a separate and independent test set would require redoing the entire analyses, this is obviously not practical or necessary. Moreover, the authors later evaluate the model on additional experimental data, which partially fills the role of independent test data. However, the authors could at least state that the results from the mean cross-validation metrics may be slightly overoptimistic compared to what one would see for truly independent data.**

Answer: Thank you for your reminder and suggestions. We have added the following sentence to the discussion section: "Additionally, although CataPro performs well on the unbiased ten-fold cross-validation dataset, it may still encounter challenges when applied to highly unfamiliar proteins, reactions, or other independent test sets.", and highlighted it in blue.

Reviewer response: The authors' response does not fully address my concern. While what they have added to the manuscript is correct, it does not address the challenges of using a 10-fold CV as the sole evaluation procedure. The challenges encountered when applied to highly unfamiliar proteins, reactions, or other independent test sets is a problem that models typically independently of the evaluation procedure. I suggest that the authors explicitly state why a 10-fold CV evaluation is likely to lead to slight overfitting when the same 10 folds are also used for hyperparameter optimization (see my original concern for more information).

- 2. When CataPro and UniKP are trained on the same data for kcat and KM prediction, the differences in model performance are small (Figure 2h). This suggests that the main advancement of CataPro for kcat and KM prediction lies in the design of the datasets rather than in a novel model architecture. This should be emphasized in the manuscript.**

Answer: Fig. 2h shows the performance of CataPro (extra tree) and UniKP on both unbiased and biased datasets, where CataPro (extra tree) means replacing linear layer with extra tree in CataPro. Since the only difference between UniKP and CataPro (extra tree) lies in the representation of the substrate, the superiority of CataPro's feature combination can be seen in Fig. 2h, although the advantage is limited, as you mentioned. Furthermore, Fig. 2h provides a more intuitive illustration of the performance differences of the models on biased and unbiased datasets. Here, we urge researchers to focus on the model's performance on unbiased datasets

to enable a more objective evaluation. In accordance with your suggestion, we have enhanced the discussion in **Section 2.2** in the revised manuscript and highlighted it in blue.

Reviewer response: What the authors have added to the manuscript is the opposite of what I aimed at with my concern. They now added that the “*difference between CataPro (extra tree) and UniKP demonstrates that the features used by CataPro are more effective than those used by UniKP*”. However, when we look at Figure 2h (upper panel for kcat), the difference between CataPro and UniKP on the biased dataset for PCC and SCC is within a range of 0.005 and for the unbiased dataset it looks like the differences are within a range of 0.02. These differences might not even be statistically significant. I repeat my prior statement: “*This suggests that the main advancement of CataPro for kcat and KM prediction lies in the design of the datasets rather than in a novel model architecture*” This is not a bad finding since a clean dataset might be even more important than some tweaking to the architecture and representations, but I still argue this should be clearly stated in the manuscript.

- 3. CataPro does not use any reaction information. The turnover number kcat depends on the whole chemical reaction catalyzed by an enzyme and not only on one substrate. However, CataPro only uses information about one of the substrates that is part of the reaction. This limitation is not mentioned or discussed anywhere in the manuscript. Why is the entire reaction not encoded? Is this limitation important for the performance of the model? How does one choose a single substrate in the case of reactions with multiple reactions?**

Answer: Thank you for your comments. We agree that reaction information is an important factor. We ultimately do not use reaction information, mainly due to the following concerns: (1) Collecting reaction information introduces additional conditions and constraints, which will reduce the sample size of the final dataset. (2) With limited sample size, it will be difficult to support the model in achieving additional functionalities, such as the ability to rank mutants.

When predicting enzyme kinetic parameters for reactions involving multiple substrates, we can select a key substrate as input. In fact, in the BRENDA database, some entries may omit one of the two substrates, and the omitted substrate is typically a common, low-molecular-weight molecule, such as water, oxygen, or metal ions. As discussed in **Section 2.7**, 4-vinylguaiacol is evidently a key substrate in the reaction.

Reviewer response: I disagree: The authors write “*some [BRENDA] entries may omit one of the two substrates, and the omitted substrate is typically a common, low-molecular-weight molecule, such as water, oxygen, or metal ions.*”, but this is not the case. In multi-substrate reactions, to measure kcat, the concentration of all but one substrate is kept fixed (and high) and the concentration of the remaining substrate is increased. BRENDA lists this substrate for which the concentration is increased. This may or may not be the key substrate. For example, if we look at the following BRENDA entry ([https://www.brenda-enzymes.org/enzyme.php?ecno=1.1.1.10#TURNOVER%20NUMBER%20\[1/s\]](https://www.brenda-enzymes.org/enzyme.php?ecno=1.1.1.10#TURNOVER%20NUMBER%20[1/s])), we see kcat values for the main substrate and for the co-factors. Thus, the CataPro dataset has

many substrate entries where the substrate is not the main/key substrate. This is a major limitation.

In addition, it would be possible to obtain the reaction equations for the majority of the data points with substantial additional work. I don't expect the authors to do this work, but to say that it is not possible is not accurate. The authors should clearly state this limitation and the potential problems it causes.

It is also unclear to me how to obtain *k_{cat}* predictions for new enzymatic reactions. Often it is not possible to define a key substrate, or when predictions are required for genome-scale metabolic models, it is impractical to hand-label all key substrates. It should be discussed how the model should be applied to obtain appropriate predictions.

- 4. Comparison with TurNuP and KM prediction model: The authors did not perform a comparison with the TurNuP model, which showed much better performance than DLKcat. While the TurNuP model architecture is not special, the authors put a lot of effort into data cleaning and preprocessing, such as determining the complete chemical reaction equations for all data points. Since TurNuP and CataPro were evaluated on different datasets, a direct fair comparison is not possible. However, if we consider (i) in both cases only test enzymes with a sequence identity of less than 40% compared to the training enzymes and (ii) that the PCC squared is approximately equal to the coefficient of determination R², we obtain the following comparison: CataPro PCC²=(0.497)²=0.247 vs. TurNuP R² = 0.328 Performing the same comparison for CataPro and the ProSmith Km (<https://doi.org/10.1371/journal.pcbi.1012100>, see Minor Concern 2.) prediction model for test sequences with less than 40% sequence identity, we obtain the following comparison: CataPro PCC²=(0.633)²=0.401 vs. ProSmith R² = 0.41 Can the authors comment on why the results (at least in this simplified comparison) do not show significant differences between the two prediction models? TurNuP and ProSmith_KM only provide predictions for wild-type enzymes? Could it be that both models perform similarly for wild-type enzymes?**

Answer: Thank you for your suggestion. To ensure a fair comparison with TurNuP, we trained and tested our model on the dataset used by TurNuP. Since the TurNuP dataset includes all substrates and products for each reaction, we extracted feature vectors for each substrate and product individually. For reactions with multiple substrates or products, we calculated the average of the corresponding feature vectors to serve as the final representation of the substrates or products. Then, the feature vectors of the enzyme, substrate, and product are concatenated to obtain the representation of the entire reaction (**Supplementary Fig. 9a**). Finally, a linear layer, as used in **Fig. 1b**, is applied to predict *k_{cat}*. We trained CataPro on the five-fold cross-validation dataset divided by Kroll et al., and then evaluated the resulting five models on the four subsets of the test set, which were divided based on the enzyme's maximum sequence identity to those in the training set. The results are shown in **Supplementary Fig. 9b**. An interesting observation is that one of the models (Fold-5) achieved an R² of 0.374 in the subset with the maximum sequence identity below 40%, slightly outperforming 0.33 of TurNuP. It is important to note that the number of training samples for each sub-model of CataPro is actually four-fifths of the TurNuP training samples, as one-fifth of the samples are used for validation. Another notable observation is that, within the subset exhibiting a maximum sequence identity

of 99%-100%, the performance of the five CataPro models varies considerably. This discrepancy could be attributed to the fact that each training set is missing one-fifth of the samples from the original dataset. We discussed these results in **Section 2.3** of the revised manuscript.

Reviewer response: Concerns regarding the comparison:

(i) The CataPro model does not utilize reaction information, because no training dataset with full reaction information was assembled. However, for the comparison with TurNuP, the authors included reaction information from the TurNuP dataset. Therefore, this version of CataPro uses more information than the original CataPro version and therefore does not show the differences between TurNuP and the presented CataPro model, which does not use full reaction information.

(ii) If we average the results from the 5-fold CV, CataPro is no better than TurNuP (maybe a little worse) across the different sequence similarities. However, the authors have simply picked the only one of the 5 folds where the model performs better (on the 0-40% similarity subset) on the test set, and they only report the result of that specific fold. This is not a fair comparison at all. If the authors are concerned about limited training set size, they can do the same as TurNuP and do hyperparameter optimisation on the 5-fold CV, pick a set of hyperparameters, and then train the model on the entire training set with the pre-selected hyperparameters.

(iii) A more general comment about all comparisons: A prediction model is defined not only by its model architecture, but also strongly by its training data and training process. However, the authors do not make these distinctions. For example, here the authors make a comparison between the TurNuP and CataPro architectures on the TurNuP dataset, which means that this comparison does not show the performance of the CataPro model presented in this study, which was trained on different data. I think it is important to make this distinction clear in the manuscript.

ProSmith is also an interesting work. The similar performance of ProSmith_KM and CataPro_KM based on your rough comparison might also suggest that the differences among current models for Km prediction are perhaps not significant, as shown in **Fig. 2**, at least smaller than the differences in kcat prediction.

Reviewer response: The authors do not discuss this equal performance between ProSmith_Km and CataPro at all in the manuscript. As there is already a KM prediction model with similar prediction capabilities, this should be discussed.

5. Analysis in section 2.4:

If I understand correctly, the authors evaluated the performance for the mutant kcat prediction for different enzyme families simultaneously. However, this can potentially lead to an overly optimistic PCC as the following simple example demonstrates: Suppose that we have three wildtype enzymes, W1, W2 & W3, with kcat values of 1/s, 100/s, and 1000/s, respectively. We now want to predict the kcat value for three mutants of these wildtypes, M1, M2 & M3, with true kcat values that are significantly different from the corresponding wildtype kcat values: 0.1/s, 130/s and 1300/s, respectively. If we have a

model that simply predicts the kcat values of the wild-type enzymes and thus predicts values close to 1/s, 100/s and 1000/s for the mutants, the model clearly cannot predict the effect of the mutations on the kcat value. However, if we compare the raw predicted values with the raw true values for the mutants, we still obtain a very high PCC value due to the large scale of the kcat values, and it appears as though we have an almost perfect mutant prediction model (see figure below). The accuracy of identifying advantageous mutations could also be slightly biased if we assume that most mutants lower the kcat value. A more intuitive and unbiased measure could be to compute for each mutant the difference compared to the corresponding wildtype kcat value and then comparing predicted differences from true differences. Such delta values as computed in <https://doi.org/10.1093/biomet/bpae061> can then be compared across different enzymes.

Answer: In fact, what we are evaluating here is the model's ability to rank mutants for each reaction (with the same UniProt ID and SMILES). For example, in **Fig. 4a-c**, we first calculated the SCC achieved by the model for each reaction, and then presented the SCC results across all reactions in the form of a box plot. Therefore, the method you mentioned is actually consistent with the approach used in **Fig. 4a-c** of our manuscript. **Fig. 4d-f** demonstrates the model's ability to predict advantageous and disadvantageous mutants. Taking **Fig. 4a** as an example, advantageous mutants are defined as those where the real kcat value is higher than that of the wild-type. If the model's predicted kcat for this mutant is greater than that for the wild-type, it indicates a successful prediction. We calculated the accuracy of the model for each reaction (with the same UniProt ID-Smiles) and presented the model's performance across all reactions in the form of a box plot. Therefore, these two metrics (SCC and Accuracy) can provide a more intuitive assessment of the model's ability to rank mutants for the specific reaction.

Reviewer response: The ranking capabilities of the resulting models for this difficult prediction task do not appear to be very high. In Figure 4a-c, the average SCC for $N \geq 30$ seems to be ≤ 0.2 , i.e. even if more than 30 mutants for the same enzyme were in the training set, the model cannot accurately rank unseen mutants. I was therefore surprised to see an accuracy of 70% in Figure 4d-f. I think it is likely that there is a bias in the dataset where, for example, for one enzyme, many of the reported mutant values are all higher or lower than the wildtype kcat values. To test if such a bias exists and if the model can actually predict something other than a bias, the authors could normalise the predictions and the true values by subtracting the mean of all values of the same enzyme-SMILES pair in the training set. If the model only learns to predict an average bias for all mutants of a particular enzyme-SMILES pair, the predictions will be close to that average.

6. How did the authors handle multimer enzymes with multiple different UniProt IDs? Which of the UniProt IDs was chosen in those cases?

Answer: During data collection, we excluded samples with multiple UniProt IDs.

Reviewer response: If I am not mistaken, this is not mentioned in the manuscript, but this is important information that should be included.

Minor concerns

1. **“The SMILES string of the substrate was downloaded from PubChem[35] on the substrate name”. How many substrates could not be mapped to SMILES?**

Answer: Among all the kcat and Km entries collected from the BRENDA and SABIO-RK databases, a total of 30,691 unique substrate names were included, but 17,306 of these could not be assigned SMILES.

Reviewer response: This information should be included in the manuscript.

2. **SMILES strings were used to represent small molecules? Were SMILES strings standardized, for example to canonical SMILES?**

Answer: Yes, small molecules are represented by canonical SMILES.

Reviewer response: This information should be included in the manuscript.